

# Exploration of the influence of environmental conditions on secondary organic aerosol formation and organic species properties using explicit simulations: development of the VBS-GECKO parameterization

Victor Lannuque[1,2,3], Marie Camredon[1], Florian Couvidat[2], Alma Hodzic[4,5], Richard Valorso[1], Sasha Madronich[4], Bertrand Bessagnet[2] and Bernard Aumont[1]

[1] LISA, UMR CNRS 7583, IPSL, Université Paris Est Créteil and Université Paris Diderot, 94010 Créteil Cedex, France.
[2] INERIS, National Institute for Industrial Environment and Risks, Parc Technologique ALATA, 60550 Verneuil-en-Halatte, France.
[3] Agence de l'Environnement et de la Maîtrise de l'Energie, 20 avenue du Grésillé - BP 90406, 49004 Angers Cedex 01, France.
[4] National Center for Atmospheric Research, Boulder, Colorado, USA.
[5] Laboratoire d'Aérologie, Observatoire Midi-Pyrénées, Université Paul Sabatier, CNRS, Toulouse, France.

*Correspondence to*: Marie Camredon (marie.camredon@lisa.u-pec.fr) and Bernard Aumont (bernard.aumont@lisa.u-pec.fr)

**Abstract.** Atmospheric chambers have been widely used to study secondary organic aerosol (SOA) properties and formation from various precursors under different controlled environmental conditions and to develop parameterization to represent SOA formation in chemical-transport models (CTM). Chamber experiments are however limited in number, performed under conditions that differ from the atmosphere and can be subject to potential artifacts from chamber walls. Here the Generator for Explicit Chemistry and Kinetics of Organics in the Atmosphere (GECKO-A) modelling tool has been used in a box model under various environmental conditions to (i) explore the sensitivity of SOA formation and properties to changes on physical and chemical conditions and (ii) to develop a Volatility Basis Set type parameterization. The set of parent hydrocarbons includes n-alkanes and 1-alkenes with 10, 14, 18, 22, and 26 carbon atoms, α-pinene, β-pinene and limonene, benzene, toluene, o-xylene, m-xylene and p-xylene. Simulated SOA yields and their dependences on the precursor structure, organic aerosol load, temperature and NOx levels are consistent with the literature. GECKO-A was used to explore the distribution of molar mass, vaporization enthalpy, OH reaction rate and Henry's law coefficient of the millions of secondary organic compounds formed during the oxidation of the different precursors and under various conditions. From these explicit simulations, a VBS-GECKO parameterization designed to be implemented in 3D air quality models has been tuned to represent SOA formation from the 18 precursors using GECKO-A as a reference. Its evaluation shows that VBS-GECKO captures the dynamic of SOA formation for a large range of conditions with a mean relative error on organic aerosol mass temporal evolution lesser than 20% compared to explicit simulations. The optimization procedure has been automated to facilitate the update of the VBS-GECKO on the basis of the future GECKO-A versions, its extension to other precursors and/or its modification to carry additional information.



# 1 Introduction

Fine particulate matter impacts visibility (e.g. Han et al., 2012), human health (e.g. Lim et al., 2012; Malley et al., 2017) and climate (e.g. Boucher et al., 2013). A large fraction of fine particles is organic, representing between 20 and 90 % of the total mass (e.g. Jimenez et al., 2009). This organic fraction can be directly emitted into the atmosphere (primary organic aerosol, POA) or formed by gas/particle partitioning of low volatility species produced during the oxidation of gaseous organic compounds (secondary organic aerosol, SOA) (e.g. Mader et al., 1952; O'Brien et al., 1975; Grosjean, 1992). The secondary origin of organic aerosol dominates the primary fraction in most environments (e.g. Gelencsér et al., 2007). SOA are composed of a large number of compounds having various chemical structures and spanning therefore a wide range of physico-chemical properties (reactivity, volatility, molar mass, solubility, optical absorption). The amount of SOA formed from a gaseous precursor depends mainly on its structure (carbon chain length, degree of unsaturation, number and type of functional groups...), and on environmental conditions that influence (i) the concentration and the structure of organic compounds produced during gas phase oxidation (oxidant concentrations and NOx levels, temperature, photolysis, humidity...) and (ii) their partitioning between the gas and the condensed phase (pre-existing organic aerosol mass ($C_{oa}$), temperature...) (e.g. Kroll and Seinfeld, 2008).

SOA formation is represented in chemical-transport models (CTMs) using simplified parameterizations. These parameterizations describe the formation of secondary organic species from the oxidation of gaseous precursors and theirs consumptions through the use of surrogate species having different volatilities (e.g. Odum et al., 1996; Pun et al., 2002, 2003 and 2006; Tsigaridis et al., 2003; Couvidat et al., 2012; Couvidat et al., 2015), sometimes discretized into decadal volatility bins (e.g. Robinson et al., 2007; Donahue et al., 2006 and 2012; Jimenez et al., 2009; Cappa and Wilson 2012; Zhang and Seinfeld, 2012). The gas-particle partitioning of a surrogate species or a given volatility bin is usually described considering an absorption process according to the theory described by Pankow (1994). Comparisons with field observations show that current CTMs have difficulties in reproducing the spatial and temporal variations of PM2.5 mass concentrations (e.g. DeGouw et al., 2005; Johnson et al., 2006; Volkamer et al., 2006; Heald et al., 2005; Hodzic et al., 2010; Shrivastava et al., 2011; Couvidat et al., 2012; Solazzo et al., 2012; Im et al., 2015), mainly because of the difficulties to represent the secondary organic fraction of the aerosol (Solazzo et al., 2012).

SOA parameterizations have typically been developed and optimized based on atmospheric chamber data. Atmospheric chambers have been widely used to study SOA formation from various precursors under different controlled environmental conditions. Experiments, being limited in number, do not cover the diversity of atmospheric chemical and physical conditions that influences SOA formation. Smog chamber experiments are also usually performed under conditions that differ from the atmosphere (high level of oxidants and precursors, different light spectrum and intensity, low humidity, short times of ageing...). In addition, SOA formation experiments can be subject to potential artefacts from chamber wall surfaces, such as heterogeneous reactions or aerosol and vapour wall losses (e.g. Mc Murry and Grosjean, 1985; Matsunaga and Ziemann, 2010; La et al., 2016).



The objectives of this study are (i) to explore the influence of environmental conditions on SOA formation and the distribution of physico-chemical properties (volatility, enthalpy of vaporization, molar mass, OH rate constant and water solubility) of the evolving molecular mixture and (ii) to optimize a Volatility Basis Set (VBS) type parameterisation (e.g. Donahue et al., 2006) on the basis of a deterministic and explicit modelling of SOA formation. Deterministic and explicit

simulations allow (i) taking into account the influence of various environmental conditions encountered in the atmosphere, (ii) simulating SOA formation under representative concentration regimes and timescales and (iii) accessing detailed properties of organic compounds in both gas and condensed phases. Nearly-explicit representation of SOA formation under various environmental conditions is possible with GECKO-A (Generator for Explicit Chemistry and Kinetics of Organics in the Atmosphere) (Aumont et al., 2005; Camredon et al., 2007). The environmental scenarios and the explicit representation

of SOA formation are described in section 2. Explicit simulations are used to explore the influence of environmental conditions on SOA formation and the distribution of the physico-chemical properties of species produced during organic compound oxidation in section 3. A VBS type parameterization is optimized and evaluated using explicit simulations as a benchmark in section 4.

## 2 Explicit simulations of SOA formation

### 2.1 The environmental scenarios

Scenarios were developed in a box model to represent the variability of SOA formation within the range of environmental conditions encountered in the atmosphere (temperature, relative humidity, solar zenith angle, aerosol mass concentration and chemical concentrations). Scenarios were built with various NOx concentrations, to cover chemical regimes from low NOx to high NOx conditions (Table 1).

To examine the sensitivity of SOA formation to each parameter, chemical and physical conditions were fixed in each scenario. Physical parameters set in the simulations were temperature (270 and 298 K), relative humidity (70%), zenith angle (30, 50 and 70°) and $C_{oa}$ (0.1, 1 and 10 µg m$^{-3}$). The chemistry considered was the reactivity of inorganic species (i.e. the Ox/NOx/HOx chemistry), and the oxidation of CO and CH$_4$. The influence of non-methane volatile organic compound (VOC) oxidation on the HOx and NOx cycles was included using a surrogate species. The OH lifetime with respect to

reaction with the surrogate VOC was set to 1 s, in the range of what is observed in situ (Lou et al., 2010; Hansen et al., 2014). The oxidation of this surrogate VOC with OH leads to the formation of a surrogate peroxy radical, with a chemistry assumed to be similar to CH$_3$O$_2$. Dry deposition was considered for O$_3$, H$_2$O$_2$ and HNO$_3$ (with deposition rates of $2.0\times10^{-4}$, $2.5\times10^{-4}$ and $2.0\times10^{-3}$ s$^{-1}$ respectively). Chemical concentrations were fixed for CO (120 ppb), methane (1750 ppb), O$_3$ (10, 40 and 70 ppb) and NOx (ranging from $10^{-2}$ to $10^2$ ppb). Photolysis frequencies were computed for each fixed zenith angle at

the surface using the TUV model (Madronich and Flocke, 1998) with rules for organic chromophores described in Aumont et al. (2005). Each simulation was run in a box model until the OH, HO$_2$, NO and NO$_2$ stationary state concentrations are





reached. These stationary states were used as inputs for our SOA formation scenarios. In each scenario, the initial concentration of a given SOA precursor is set to a low enough value (10 pptC) to not affect significantly both the stationary state conditions of the simulations and the prescribed amount of pre-existing aerosol ($C_{oa}$).

Figure 1 shows the simulated stationary state concentrations of OH and $HO_2$ as a function of NOx concentrations for the

various scenarios. Simulated evolutions and concentrations of OH and $HO_2$ with NOx are in agreement with the observations (e.g. Stone et al., 2012). The distribution of secondary species depends on the evolution of $RO_2$ radicals. The branching ratio of $RO_2$ reacting with NO, called α, is often used to characterize the chemical environment (e.g. Lane et al., 2008). In the scenarios, the loss of $RO_2$ with $CH_3O_2$ (the major $RO_2$ radical) represents less than 10 %. The α ratio was therefore calculated here without considering the $RO_2 + RO_2$ reactions as:

$$\alpha = \frac{k_{RO_2+NO}[NO]}{k_{RO_2+NO}[NO]+k_{RO_2+HO_2}[HO_2]} ,$$
(1)

where $k_{RO2+NO}$ and $k_{RO2+HO2}$ are the rate constants for the reactions of the peroxy radicals with NO, and $HO_2$ respectively and [NO] and [$HO_2$] the concentration of the different radicals. Figure 1 also presents the variation of α with NOx. Rate constants of $RO_2$ with NO and $HO_2$ used to compute α are calculated at 298 K: $k_{RO_2+NO} = 9.0 \times 10^{-12}$ cm$^3$ molec$^{-1}$ s$^{-1}$ (Jenkin et al., 1997) and $k_{RO_2+HO2} = 2.2 \times 10^{-11}$ cm$^3$ molec$^{-1}$ s$^{-1}$ (e.g. Boyd et al., 2003, assuming a large carbon skeleton for $RO_2$). Figure 1 shows

that most of the $RO_2$ reacts with NO for NOx concentrations higher than 1 ppb. The NOx and HOx concentrations used in the various scenarios are consistent with typical chemical characteristics of low- to high-$NO_X$ environments. These stationary state scenarios are representative of the evolution of the tropospheric Ox/NOx/HOx chemical system. They were used here to explore explicitly the formation of SOA under the ranges of NOx levels, oxidant concentrations, temperature, solar radiation and pre-existing organic aerosol mass encountered in the troposphere.

**2.2 The explicit representation of SOA formation**

Aromatic and long chain aliphatic hydrocarbons emitted by anthropogenic activities, and isoprene and terpenic compounds emitted by biogenic sources, are known to be major SOA precursors (e.g. Carlton et al, 2009; Kroll and Seinfeld, 2008; Nordin et al., 2013; Tsigaridis et al., 2014; Zhao et al., 2016). Explicit oxidation chemical schemes were generated with the GECKO-A modelling tool for 5 linear alkanes (decane, tetradecane, octodecane, docosane and hexacosane), 5 linear 1-

alkenes (decene, tetradecene, octodecene, docosene and hexacosene), 5 aromatic compounds (benzene, toluene, m-xylene, o-xylene and p-xylene) and 3 terpenic compounds (α-pinene, β-pinene and limonene). The GECKO-A modelling tool does not handle the polycyclic aromatic structures and the current oxidation protocol is not up to date to represent SOA formation from isoprene oxidation. SOA formation from polycyclic aromatic hydrocarbons and isoprene oxidation has thus not been studied here.

The GECKO-A tool is a computer program that automatically writes chemical schemes on the basis of a prescribed protocol. Elementary data are taken from laboratory studies if available and structure/property relationships if not. The protocol implemented in GECKO-A is described by Aumont et al. (2005), with chemistry updates for gas phase oxidation performed





by Valorso et al. (2011) and Aumont et al. (2012, 2013). The evolution of alkoxy radicals through decomposition and isomerisation is estimated using the structure/property relationship developed by Vereecken and Peeters (2009, 2010), as described by La et al. (2016). In the current version, GECKO-A generates explicit chemical schemes for non-aromatic species and uses the Master Chemical Mechanism (MCM, version 3.3.1) to describe the chemistry of species including an

aromatic structure (Jenkin et al., 2003; Bloss et al., 2005).

The explicit description of the oxidation of organic precursors involves millions of secondary species and reactions. Explicit chemical schemes cannot be solved for precursors having more than 8 atoms of carbon, even in a box model. Simplifications are therefore needed. In this study, the gaseous reactivity of species having a saturation vapour pressure lower than $10^{-13}$ atm is not considered as these species are dominantly in the condensed phase. Lumping of position isomers is also allowed if the

production yield of a species is lower than $10^{-3}$ (Valorso et al., 2011). $RO_2$ mainly reacts with NO, $HO_2$ and other $RO_2$. In our scenario, the $RO_2$ family is dominated by $CH_3O_2$. Among the self $RO_2$ reactions, only the $RO_2+CH_3O_2$ are considered when chemical schemes are generated. Furthermore, writing of the oxidation scheme is stopped after formation of the $15^{th}$ generation stable products.

The gas/particle partitioning of each stable organic compound was implemented as described in Camredon et al. (2007).

Phase partitioning is described by an absorption process following the Raoult's law (e.g. Pankow, 1994):

$$C_p^i = \frac{R \, T \, C_{OA}}{\gamma_i \, P_i^{sat} \, \overline{M}_{OA}} \times C_g^i \,, \qquad (2)$$

where $C_p^i$ and $C_g^i$ are the concentration of the species i in the particulate and the gas phase respectively, R is the gas constant, T the temperature, $\overline{M}_{OA}$ the mean molar weight of the organic aerosol (250 g mol$^{-1}$), $P_i^{sat}$ the saturation vapour pressure of the species i and $\gamma_i$ the activity coefficient of the species i in the condensed phase. The condensed phase is considered

homogeneous, ideal ($\gamma_i$=1), inert and at equilibrium with the gas phase. Saturation vapour pressure of each organic compound is estimated using the Nannoolal method (Nannoolal et al., 2004; Nannoolal et al., 2008). Equation (1) is solved for the set of organic species using a simple iterative method (Pankow, 1994). For the gas phase oxidation, time integration is solved using the 2-step solver (Verwer and Vanloon, 1994; Verwer et al., 1996).

## 3 Exploration of SOA formation and organic species properties

### 3.1 Carbon distribution and SOA formation

Figure 2 presents the temporal evolution of the carbon during the oxidation of a precursor in the scenario performed at a temperature of 298 K, a zenith angle of 50°, a $C_{oa}$ of 1 µg.m$^{-3}$, 40 ppb of ozone and a α of 90%. This scenario is called hereafter the "reference" scenario. Examples are shown for o-xylene, α-pinene and n-octadecane oxidation. Under these average conditions, the precursors are oxidized in 10, 6 and 2.5 hours for o-xylene, n-octadecane and α-pinene respectively.

The carbon is first dominantly in the form of secondary gaseous organic compounds (around 55, 85 and 40% at the gaseous maximum for o-xylene, α-pinene and n-octadecane respectively). For o-xylene these gas-phase intermediates are mainly



transformed into CO and $CO_2$, with less than 1% of the carbon present as secondary organic aerosol. After 5 days of oxidation, the carbon mainly ends in the form of CO and $CO_2$ (~85 %), the remaining fraction being mostly gas phase organic carbon. The complete oxidation of α-pinene into CO and $CO_2$ takes longer than for o-xylene. During the oxidation, up to 7% of the carbon is in SOA. After 5 days, the carbon initially present as α-pinene mainly ends in the form of CO and

$CO_2$ (~85 %); the remaining fraction of the carbon is gaseous at 10% and condensed at 5%. For n-octadecane, the full oxidation of the carbon into CO and $CO_2$ is slow. Gas-phase intermediates produce dominantly low volatility species which transfer to the condensed phase. A maximum carbon fraction of around 70% is in the particle phase after 10 hours of oxidation. The carbon fraction in SOA is slowly transferred back to the gas phase when gaseous oxidation proceeds, leading ultimately to the formation of CO and $CO_2$. The oxidation of the carbon into CO and $CO_2$ occurs only by gas phase oxidation

here as the particle phase is represented as inert in this model configuration. After 5 days of oxidation, the carbon is mainly in the aerosol (~62%) and in the form of CO and $CO_2$ (~38%).

**3.2 Influence of environmental parameters on SOA formation**

Figure 3.a shows the evolution of the maximum SOA yield ($Y_{max}$, calculated as the mass ratio of the produced SOA to the reacted precursor quantity) as a function of the number of carbons in the parent compound. Results are shown for the

"reference" scenario; the maximum SOA yield is noted $Y_{max}^{ref}$. For a given 1-alkane or 1-alkene series, simulated SOA yield growths with the length of the precursor's carbon skeleton. This behaviour as well as the simulated yields are consistent with observations from chamber experiments during n-alkane and 1-alkene oxidation at ambient temperature and high-NOx (e.g. Lim and Ziemann, 2009; Matsunaga et al., 2009). $Y_{max}$ reaches a plateau of about 1.2 for precursor's chain lengths higher than 18 carbons, independently of their carbon number. These precursors lead to the formation of first generation compounds

being dominantly in the condensed phase (e.g. Aumont et al., 2012). For aromatic compounds, simulated $Y_{max}$ decreases with increasing number of methyl groups. This $Y_{max}$ evolution with the aromatic precursor's structure is consistent with experimental trends observed in chambers (e.g. Ng et al., 2007; Li et al., 2016). For terpenes, simulated $Y_{max}$ increases in the following precursor's order: α-pinene, β-pinene and limonene, in agreement with observed behaviours (e.g. Zhao et al., 2015).

The influence of temperature and pre-existing organic aerosol on $Y_{max}$ is presented in Figure 3.b and 3.c. Results are shown for a scenario in which one environmental parameter is changed from the "reference" scenario: the temperature from 298 to 270 K or the $C_{oa}$ from 1 to 0.1 or 10 µg m$^{-3}$ (the maximum SOA yields are noted hereafter $Y_{max}^{270K}$, $Y_{max}^{0.1µg.m^{-3}}$ and $Y_{max}^{10µg.m^{-3}}$ respectively). As expected, simulation results show that aerosol yields increase when $C_{OA}$ increases (e.g. Odum et al., 1996) and when the temperature decreases (e.g. Takekawa et al., 2003). In both cases, the highest sensitivity is simulated

for species bearing between 6 and 14 carbon atoms. For these precursors, the secondary organic species contributing to SOA are semi-volatile and their partitioning depends largely on the temperature and $C_{oa}$. A low sensitivity of $Y_{max}$ is simulated for



the species with the longest carbon skeleton (i.e. with $n_C > 18$). These low volatility precursors rapidly form low volatility compounds that partition dominantly to the condensed phase, regardless of environmental conditions.

The influence of NOx on $Y_{max}$ is presented in Figure 4. Results are shown for the "reference" scenario ($\alpha = 90\%$) and scenarios into which the $\alpha$ value is changed from the "reference" scenario to 0, 10, 50 and 100%. Simulated results show that

SOA yields generally decreases with the increase of NOx. These trends are in agreement with observations (e.g. Ng et al., 2007, Donahue et al., 2005). Again, a weak sensitivity to NOx is simulated for largest chain length species (i.e. with $n_C > 18$). These larger chain length compounds produce first oxidation products able to partition dominantly in the condensed phase at all NOx levels. The explicit simulations reproduce the observed tendencies of SOA yields with the precursor's structure, NOx levels, temperature and pre-existing organic aerosol mass. These agreements support their use to explore the

physico-chemical properties of organic compounds and as a reference for the development of reduced chemical schemes for SOA formation.

### 3.3 Physico-chemical properties of organic compounds

The explicit simulations were used to explore the distribution of organic species physico-chemical properties that influence organic aerosol formation, in particular:

(1) the molar mass (Mw), needed to convert the aerosol concentration of the species from molecular to mass units;

(2) the volatility (i.e. saturation vapour pressure) that influences directly the gas/aerosol partitioning of a species. The saturation vapour pressure at a given temperature is calculated in GECKO-A using the Clausius-Clapeyron expression with $P^{sat}$ at 298K and vaporization enthalpy ($\Delta H_{vap}$) estimated from the Nannoolal et al. method (2004, 2008);

(3) the solubility (i.e. Henry's law coefficient) that influences the dry deposition of organic gases but also potentially its

gas/aerosol partitioning for water soluble species. The effective Henry's law coefficient ($H^{eff}$) at 298K is calculated in GECKO-A with the GROHME method (Raventos-Duran et al., 2010);

(4) the reaction rate with OH ($k_{OH}$) that drives the chemical lifetime of organic species. The OH reaction rate at 298 K of each organic compound is estimated in GECKO-A using the Kwok and Atkinson (1995) structure activity relationship with updates described by Aumont et al. (2012) if no experimental data is available.

Figure 5 presents the simulated distribution of (a) molar mass, (b) vaporisation enthalpy, (c) OH reactivity and (d) solubility ($H^{eff}_{298K}$) as a function of the volatility ($P^{sat}_{298K}$) of organic species. Results are presented for a time step close to the maximum of secondary organic compound, after 6 hours of oxidation, for the "reference" scenario. Examples are shown for o-xylene, $\alpha$-pinene and n-octadecane oxidation. Thousands of secondary organic species spanning several orders of magnitudes in properties are formed during the oxidation of a given precursor. The volatility distributions range from $10^{-15}$ to

1 atm, independently of the precursor structure. Molar mass increases with the decrease of the volatility of organic compounds, varying between around 50 to 400 g.mol$^{-1}$ for the highest and lowest volatile species. Simulated results show that species formed at high NOx have a molar mass around 100 g.mol$^{-1}$ higher than at low NOx (see Figure S1 in



Supplementary Material). Explicit simulation shows a $\Delta H_{vap}$ increase of 12.5 kJ.mol$^{-1}$ per logarithm unit decrease of $P^{sat}_{298K}$, independently to NOx levels (see Figure S1 in Supplementary Material). Rate constant of organic species with OH at 298 K vary between $10^{-13}$ and $10^{-10}$ molec$^{-1}$.cm$^3$.s$^{-1}$. Simulation do not show a clear trend of $k_{OH}$ with the volatility of the species, neither with the level of NOx, $k_{OH}$ being scattered over 3 orders of magnitude for a given $P^{sat}$ and $\alpha$ ratio. Effective Henry's

law constant increases with the decrease of organic compound volatility, varying between around 1 and $10^{16}$ mol.L$^{-1}$.atm$^{-1}$ for the highest and lowest volatile species. NOx levels do not have a significant influence on the $H^{eff}$ distribution of species with the volatility (see Figure S1 in Supplementary Material).

## 4 Development and evaluation of the VBS-GECKO parameterization

### 4.1 Structure of the parameterization

A Volatility Basis Set type parameterization (e.g. Donahue et al, 2006) was developed on the basis of the explicit GECKO-A simulations. The parameterization, called VBS-GECKO, lumps the secondary organic compounds (SOCs) into bins of saturation vapour pressures at 298 K ($P^{sat}_{298K}$) and considers:

- The formation of n volatility bins ($VB_{k,i}$ with i varying from 1 to n) from the gas-phase oxidation for each precursor k ($precu_k$). The $VB_{k,i}$ are considered to be formed from OH oxidation for all the precursors, and also from $O_3$ and $NO_3$

oxidation for alkenes and terpenes. The reaction rate constants for the oxidation of a given precursor k ($k_{precuk+OH}$, $k_{precuk+O3}$ and $k_{precuk+NO3}$) are taken from the GECKO-A database (see Table S1 in the Supplementary Material). Each $VB_{k,i}$ is formed with a molecular stoichiometric coefficient that depends on NOx via the $\alpha$ ratio ($a_{k,\alpha,i}$, $b_{k,\alpha,i}$ and $c_{k,\alpha,i}$ for OH, $O_3$ and $NO_3$ reaction respectively):

$$\mathbf{precu_k^{(g)} + OH \rightarrow a_{k,\alpha,1} VB_{k,1} + a_{k,\alpha,2} VB_{k,2} + ... + a_{k,\alpha,n} VB_{k,n}} \qquad \mathbf{k_{precuk+OH}}$$

$$\mathbf{precu_k^{(g)} + O_3 \rightarrow b_{k,\alpha,1} VB_{k,1} + b_{k,\alpha,2} VB_{k,2} + ... + b_{k,\alpha,n} VB_{k,n}} \qquad \mathbf{k_{precuk+O3}}$$

$$\mathbf{precu_k^{(g)} + NO_3 \rightarrow c_{k,\alpha,1} VB_{k,1} + c_{k,\alpha,2} VB_{k,2} + ... + c_{k,\alpha,n} VB_{k,n}} \qquad \mathbf{k_{precuk+NO3}}$$

- The ageing of the $VB_{k,i}$ from gaseous OH oxidation and photolysis, redistributing the matter between the various $VB_{k,i}$. The OH and photolysis gaseous ageing is considered for all the $VB_{k,i}$ except for the lowest volatility bin, the gas phase fraction of that bin being expected to be negligible under atmospheric conditions. The OH reaction rate constant, $k_{OH}$, was set to the

same value for each $VB_{k,i}$. These $VB_{k,i}$ + OH reactions lead to the formation of the $VB_{k,j}$ with a stoichiometric coefficient $d_{k,\alpha,i,j}$ depending on $\alpha$ ratio :

$$\mathbf{VB_{k,i}^{(g)} + OH \rightarrow d_{k,\alpha,i,1} VB_{k,1} + d_{k,\alpha,i,2} VB_{k,2} + ... + d_{k,\alpha,i,n} VB_{k,n}} \qquad \mathbf{\forall i \neq n} \qquad \mathbf{k_{OH}}$$

Each $VB_{k,i}$ ($i \neq n$) is photolysed with a photolysis constant being proportional to the acetone one ($J_{acetone}$). The proportionality factor, $\phi_k$, is considered to be the same for the $VB_{k,i}$ of a given precursor k. The $VB_{k,i}$ photolysis leads to a loss of matter:

$$\mathbf{VB_{k,i}^{(g)} + h\nu \rightarrow carbon\ lost} \qquad \mathbf{\forall i \neq n} \qquad \mathbf{\phi_k\ J_{acetone}}$$



- The partitioning of the precursor k and the $VB_{k,i}$ between the gas and the particle phase. The partitioning is described, as in the reference GECKO-A simulations, by an absorption process following the Raoult's law and considering a homogeneous, ideal, inert condensed phase at equilibrium with the gas phase:

$$\mathbf{precu_k^{(g)} \leftrightarrow precu_k^{(p)}}$$

$$\mathbf{VB_{k,i}^{(g)} \leftrightarrow VB_{k,i}^{(p)}}$$

## 4.2 VBS-GECKO properties

Explicit simulations were used to select the number and the range of the volatility bins and their properties (molar weight, $k_{OH}$, vaporisation enthalpy and Henry's law constant). Tests were performed to establish the best number and range of volatility bins to get a compromise between the reliability and the size of the VBS-GECKO parameterization. The

partitioning of organic species having a $P^{sat}_{298K}$ lower than $10^{-12}$ atm is almost exclusively in the particulate phase under typical atmospheric conditions. Therefore, the lower volatility bin in the VBS-GECKO parameterization lumps all species having a $P^{sat}_{298K}$ lower than $10^{-12.5}$ atm. Similarly, organic species having a $P^{sat}_{298K}$ above $10^{-6}$ atm are dominantly in the gas phase under typical atmospheric conditions and volatility bin is included for species having a $P^{sat}_{298K}$ above $10^{-5.5}$ atm. Between these thresholds, the gas/particle partitioning of a species depends sensitively on temperature and organic aerosol

load, so that a finer volatility discretization is desirable. A total of 7 volatility bins were selected for the VBS-GECKO parameterization. The boundaries of the k volatility bins i, $VB_{k,i}$, are the same for all the precursors k. Bin intervals at 298 K are shown in Table 2 and in Figure 5 by the different colours of the SOC bubbles.

The properties of the k volatility bins i, $VB_{k,i}$, are set to the same value for the various precursors k. Each of the $VB_{k,i}$ has for assigned saturation vapour pressure the central value of its logarithmic interval (see Table 2). Molar weight, vaporisation

enthalpy and Henry's law constant for each bin i are set to the mean properties of the SOCs in this volatility bin (i.e. the SOC property weighted by its concentration) during the alkanes explicit simulations (C10, C14, C18, C22 and C26). A same $k_{OH}$ value of $4.10^{-11}$ $cm^3$ $molec^{-1}$ $s^{-1}$ is used for all the $VB_{k,i}$ (except for the lowest volatility bin which is considered inert). This value is in the upper range of the SOC rate constants seen in Fig. 5 and is similar to that used earlier in VBS parameterisation (e.g. Robinson et al., 2007; Grieshop et al., 2009; Hodzic et al., 2010). Note that the performance of the VBS-GECKO

parameterization should only be weakly sensitive to the $k_{OH}$ value assigned to each $VB_{k,i}$. Indeed, compensations occur during the optimization of the $d_{k,\alpha,i,j}$ stoichiometric coefficients (i.e. an upper $k_{OH}$ value for the reactivity of the $VB_{k,i}$ will lead to high values of the $d_{k,\alpha,i,i}$ associated to the production of the reacting $VB_{k,i}$ bin and low values of the $d_{k,\alpha,i,j}$ associated to the production of the other bins). The properties assigned to each bin are summarized in Table 2.

In the published VBS type parameterizations, molar weights are usually set to a value of 250 g $mol^{-1}$ (e.g. Tsimpidi et al.,

2010; Robinson et al., 2007). This value is smaller than the mean values provided by the explicit simulations for the less volatile bins. The vaporization enthalpy used for each bin is also generally smaller in the VBS parameterization, especially for the less volatile bins (e.g. Robinson et al., 2007; Grieshop et al., 2009; Hodzic et al., 2010; Tsimpidi et al., 2010). The



difference reaches 40 kJ.mol$^{-1}$ for the less volatile bins. The vaporization enthalpies recently used by Hodzic et al. (2016) and derived from experimental data are however consistent with the VBS-GECKO values obtained here. Dry deposition of low volatility organic compounds has been generally ignored in VBS models and no Henry's law coefficient was therefore assigned to volatility bins until Hodzic et al. (2014). Hodzic et al. (2014) provided Henry's law coefficient for the various

volatility bins based on GECKO-A explicit simulations for anthropogenic and biogenic precursors. As expected, Henry's law coefficients provided in table 2 agree with most of the values given in Hodzic et al. (2014), both being derived from similar type of simulations.

### 4.3 VBS-GECKO optimization

### 4.3.1 Optimisation scenario and method

Explicit simulations were used to optimize stoichiometric coefficients and photolysis factors of the VBS-GECKO parameterization. A set of stoichiometric coefficients was optimised for each of the 18 precursor's k and for each α ratio. The entire range of α is covered by linear interpolation of the stoichiometric coefficients. Each set (k,α) of stoichiometric coefficients is optimized using the explicit simulations presented in section 2, with some reactions turned successively off in the GECKO-A chemical schemes to specifically evaluate the coefficients related to the reaction with each oxidant (OH, O$_3$,

NO$_3$) and photolysis, as described hereafter and summarized in Table 3.

For each n-alkane and aromatic k and for each α ratio, the 7 $a_{k,\alpha,i}$ (i=1,7) and 42 $d_{k,\alpha,i,j}$ (i=1,6 ; j=1,7) coefficients were optimized simultaneously using 6 explicit simulations performed with a zenith angle of 50°, 40 ppb of ozone, at 2 temperatures (270 and 298 K) and for 3 values of $C_{oa}$ (0.1, 1 and 10 µg.m$^{-3}$). For these optimizations, the photolysis reactions of secondary organic compounds were removed in both the GECKO-A and the VBS-GECKO chemical schemes.

For each 1-alkenes and terpenes k, and for each α ratio, the 7 $a_{k,\alpha,i}$ (i=1,7), the 7 $b_{k,\alpha,I}$ (i=1,7), the 7 $c_{k,\alpha,i}$ (i=1,7) and the 42 $d_{k,\alpha,i,j}$ (i=1,6 ; j=1,7) coefficients were optimized simultaneously on 11 explicit simulations and considering in each simulation the oxidation of the precursor k with a single oxidant only (either OH, O$_3$ or NO$_3$). For the simulations taking into account the precursor + OH oxidation only, the same set of 6 simulations described above for alkanes and aromatics was used, i.e. a zenith angle of 50°, 40 ppb of ozone, at 2 temperatures (270 and 298 K) and for 3 values of $C_{oa}$ (0.1, 1 and 10

µg.m$^{-3}$). For the simulations accounting only for the precursor reaction with O$_3$, a set of 3 simulations was used with 3 O$_3$ concentrations (10, 40 and 70 ppb), at 298 K and a $C_{oa}$ of 1 µg.m$^{-3}$. For the simulations accounting only for the precursor reaction with NO$_3$, a set of 2 simulations was used with 40 ppb of ozone, at 298 K, a $C_{oa}$ of 1 µg.m$^{-3}$ and for 2 NO$_3$ concentrations (0.04 and 0.4 ppt for the terpenes, 0.4 and 40 ppt for the 1-alkenes). For each oxidant case, the reactions of the precursor with the other oxidants and the photolysis reaction of secondary organic compounds were turned off in both the

GECKO-A and the VBS-GECKO chemical schemes.

The set of 49 stoichiometric coefficients for the alkanes and aromatics precursors and the 63 stoichiometric coefficients for the 1-alkenes and terpenes were optimized for 5 α ratio (0, 0.1, 0.5, 0.9 and 1). After the optimization of these coefficients,





the photolysis factor $\phi_k$ was finally optimized for each of the 18 precursors k. Each photolysis factor $\phi_k$ does not depend on α ratio and was optimized simultaneously using the 15 simulations performed for a precursor k with 3 values of zenith angle (30, 50 and 70°) for the 5 α ratio, 40 ppb of ozone, at 298 K and a $C_{oa}$ of 1 µg.m$^{-3}$. The optimization was performed on different durations depending on the lifetime of the studied precursor with OH. For the less reactive species (aromatic compounds and n-alkanes), the optimization was realized on a duration of 5 days. For 1-alkenes and terpenes, the optimization duration was 2 and 1 day respectively. The time step used for the optimization was set to 20 minutes.

The optimizations were performed minimizing the Root Mean Square error (RMSE) between the GECKO-A and VBS-GECKO simulated evolutions of the total mass concentration of each of the 7 volatility bins (particulate and gaseous summed). The RMSE was calculated on the entire duration of a set of reference simulations as:

$$\text{RMSE} = \sqrt{\frac{\sum_{i=1}^{n}\left\{\frac{\sum_{j=1}^{n_{time}}\left(m_{i,j}^{RED}-m_{i,j}^{REF}\right)^2}{n_{time}}\right\}}{n}},\tag{3}$$

where $m_{i,j}^{REF}$ and $m_{i,j}^{RED}$ are the mass of the volatility bin i at the time step number j simulated with the GECKO-A and the VBS-GECKO respectively, $n_{time}$ the number of time steps over the reference simulations and n the total number of bins. The optimization was run following a Bound Optimization by Quadratic Approximation method BObyQA (Powell M.J.D., 2009) under the R software (R Core Team, 2017). The fitted stoichiometric coefficients were bounded between 0 and 1 and the photolysis factors between 0 and 100. The iterative process to minimize the RSME stops when none of the parameters vary more than 0.2% between two successive iterations.

For each n-alkane or aromatic precursor, the 246 parameters (49 stoichiometric coefficients for each of the 5 α ratio plus one photolysis coefficient) were adjusted on a total of 45 learning simulations (6 simulations for each of the 5 α ratio + 15 simulations for the photolysis coefficient), leading to a total of 16200 simulated time steps (45 simulation of 120 hours each with 3 time steps per hour) providing distinct mass concentrations for the 7 volatility bins. Similarly, for each 1-alkene or terpene, the 316 parameters (63 stoichiometric coefficients for each of the 5 α ratio plus one photolysis coefficient) were optimized on 70 learning simulations (11 simulations for each of the 5 α ratio + 15 simulations for the photolysis coefficient) leading to a total 10080 simulated time steps for 1-alkene (70 simulations of 48 hours each with 3 time steps per hour) and 5040 for terpene (70 simulations of 24 hours each with 3 time steps per hour). The fitted parameters of the VBS-GECKO parameterization are provided in Table S1 of Supplementary Material.

### 4.3.2 Optimisation results

Figure 6 presents the temporal evolution of the mass concentration of each volatility bin (gas and particle summed) and of organic aerosol in the GECKO-A and VBS-GECKO simulations during the oxidation of a precursor. Results are shown for the simulations performed at a temperature of 298 K, a zenith angle of 50°, a $C_{oa}$ of 1 µg.m$^{-3}$, 40 ppb of ozone and a α of 90% (i.e. in the scenario previously called the "reference" scenario in Sec. 3.1). Examples are given for o-xylene, α-pinene





and n-octadecane oxidation. Note that in Fig. 6, organic species with $P^{sat}_{298K}$ values greater than the VBS-GECKO volatility bin upper boundary (i.e. $10^{-5.5}$ atm, see Figure 5) are not reported. At the beginning of the oxidation, GECKO-A explicit simulations show that first oxidation species are mainly in the highest volatility bins of the VBS-GECKO volatility range: (1) in VB1 and VB2 for o-xylene, (2) mainly in VB1, but also in VB2, VB3 and VB4, for α-pinene and (3) in VB2 and VB3 for n-octadecane. The oxidation of these first oxidation species leads to the formation of further generation species with a lower volatility, increasing the VB5, VB6 and VB7 mass concentration. As oxidation proceeds, the concentration of the volatility bins generally increases to reach a maximum before decreasing. Nevertheless, this behaviour is not observed for:

- the VB7 temporal evolution. In this version of the GECKO-A tool, VB7 has no chemical sink (see above) and the concentration of the VB7 therefore increases to reach a plateau.

- the VB2 temporal evolution of o-xylene. Two species contribute to most of this bin in the explicit simulations: a dinitrocresol and a dinitro-dimethyl-phenol. Under the conditions of this simulation, these second generation species have long lifetimes with OH and $NO_3$ (hundreds of hours) and therefore the chemical loss of VB2 cannot be observed on the timescale of the simulation.

Figure 6 shows that the structure of the VBS-GECKO parameterisation reproduces the temporal evolution of the 7 volatility bin's mass concentration during this learning simulation. The VBS-GECKO parameterisation shows however discrepancies in representing the evolution of the highest volatility bins (VB1 and VB2). The temporal evolution of organic species in VB1 and VB2 is driven by the reactivity of species having a $P^{sat}_{298K}$ values greater than $10^{-5.5}$ atm. As no volatility bin is included for these species in the VBS-GECKO, the VB1 and VB2 mass evolution is less well reproduced than the bins of lower volatility. Species in VB1 and VB2 do however not partition substantially into the condensed phase except under heavy aerosol loading and low temperature for VB2. The reliability of the optimization was evaluated comparing the temporal evolution of organic aerosol mass (which is not directly optimized during the fitting process) between GECKO-A and VBS-GECKO simulations (see Figure 6). The Relative Root Mean Square error (RRMSE) was calculated *a posteriori* for the aerosol mass concentration as:

$$\text{RRMSE} = \sqrt{\frac{\sum_{j=1}^{n_{time}}\left\{\left(\frac{M_j^{RED}-M_j^{REF}}{M_j^{REF}}\right)\right\}^2}{n_{time}}},\qquad(4)$$

with $M_j^{REF}$ and $M_j^{RED}$ are the mass concentration of organic aerosol formed at each time step j with the GECKO-A and the VBS-GECKO respectively and $n_{time}$ the total number of time steps. The temporal evolution of organic aerosol mass is well reproduced by the parameterization for o-xylene, α-pinene and n-octadecane, with a RRMSE lower than 20% (10.2%, 17.1% and 1.8% for o-xylene, α-pinene and n-octadecane respectively).

Figure 7 shows the distributions of the RRMSE for the organic aerosol mass concentration for the various learning simulations and for each precursor. For the n-alkanes and 1-alkenes, the RRMSE increases when the carbon chain length decreases. The mean RRMSE on OA mass evolutions is lesser than 5% for n-alkanes and 1-alkenes bearing more than 14



carbon atoms. It reaches 23 and 31% for n-decane and 1-decene. For terpenes, the mean of RRMSE are of 24, 23 and 17% for α-pinene, β-pinene and limonene respectively. For the aromatic precursors, the mean of RRMSE are of 23, 45, 32, 36 and 20% for benzene, toluene, o-, m- and p-xylene respectively. The VBS-GECKO parameterization reproduces SOA masses with a RRMSE lesser than 50% for more than 90% of the learning simulations.

## 4.4 VBS-GECKO evaluation

### 4.4.1 Evaluation scenario

Results of the GECKO-A simulations performed in the previous sections were used as a training dataset to fit the coefficients of the parameterization. Here, new simulations are conducted with GECKO-A to produce a validation dataset to assess VBS-GECKO. New box model scenarios were therefore designed for that purpose. These scenarios use a mixture of precursors and environmental conditions not encountered during the training phase, including variations of environmental parameters with time ($C_{oa}$, temperature, photolysis and NOx levels). Scenarios were run for 2 constant $C_{OA}$ values (2 and 20 µg m$^{-3}$), two types of meteorological conditions for temperature, relative humidity and zenith angle (a "summer" and a "winter" case, called hereafter SUM and WIN respectively) and two NOx conditions (a "high_NOx" and a "low_NOx" case, called hereafter HNOx and LNOx respectively). A total of 8 scenarios were run.

Diurnal temperature profiles were represented by a sinusoid function with a 5 K amplitude, a maximum reached at 2 pm and average values of 290 and 275 K for the SUM and WIN scenario respectively. Diurnal photolysis frequencies were computed for mid-latitude using the TUV model (Madronich and Flocke, 1998) with zenith angle variations corresponding to the 1$^{st}$ of July for the SUM case and 1$^{st}$ of January for the WIN scenario. The relative humidity was set to 60 and 85 % for the SUM and WIN scenario. For the HNOx and LNOx cases, the initial concentration of NOx was set at 50 and 0.5 ppb respectively. The OH lifetime with respect to the reaction with the surrogate VOC was again set to 1 s. Chemical concentrations were also fixed for CO, methane and $O_3$ at 120, 1750 and 40 ppb respectively. The evaluation scenarios started at 12 pm and lasted 5 days. Simulated evolutions of NO, $NO_2$, $NO_3$, OH, $HO_2$ and $O_3$ concentrations for the HNOx_WIN, HNOx_SUM, LNOX_WIN and LNOx_SUM scenarios are presented in Fig. S2 to Fig. S5 in Supplementary Material.

For each scenario, 3 mixtures of precursors summing up to an initial concentration of 10 pptC were considered: a "biogenic", an "intermediate" and an "anthropogenic" case (called hereafter BIO, INT and ANT respectively). In the BIO case, biogenic compounds (i.e. α-pinene, β-pinene and limonene) represent 90% in carbon of the initial mixture and anthropogenic compounds (i.e. the 5 alkanes, the 5 alkenes and the 5 aromatic compounds) the remaining 10%. In the INT and ANT cases, the biogenic / anthropogenic ratio in carbon is 50% / 50% and 10% / 90% respectively. In the biogenic or anthropogenic family, each species was arbitrary to the same level (in pptC unit).

A total of 24 simulations (8 environmental scenarios using each 3 sets of initial mixture of precursors) were run with GECKO-A chemical scheme and VBS-GECKO parameterization.





### 4.4.2 Evaluation results

Figure 8 shows the temporal evolution of the organic aerosol (OA) mass concentrations formed according to the GECKO-A and the VBS-GECKO, for the different evaluation scenarios performed with a $C_{oa}$ of 2 µg m$^{-3}$. Temporal evolutions simulated with a $C_{oa}$ of 20 µg m$^{-3}$ are presented in Fig. S6 in Supplementary Material. Figure 9 presents the distribution of

the relative errors on the OA mass concentrations for the 24 evaluation scenarios. The relative error (RE) is calculated at each time step i as:

$$RE_i = \frac{(M_i^{RED} - M_i^{REF})}{M_i^{REF}}, \qquad (5)$$

with $M^{REF}_i$ and $M^{RED}_i$ are the mass concentration of organic aerosol formed at each time step i with the GECKO-A and the VBS-GECKO respectively.

Reference GECKO-A simulations show generally (1) a fast increase of organic aerosol mass concentration during the first 10h of oxidation, (2) a clear diurnal cycle of OA concentration linked to the temperature variations, with a larger amplitude for the WIN than for the SUM scenarios, (3) higher simulated OA concentrations for the LNOx scenarios with maximum concentration obtained during the summer, (4) a slight increase of OA concentrations with the increase of anthropogenic precursors in the HNOX scenarios and with the increase of biogenic precursors in the LNOX scenarios. These trends are

well represented by the VBS-GECKO parameterization as well as the orders of magnitudes of the OA concentrations (mean errors between +/-20%). Low errors are simulated for the ANT mixture (with $|RE_i| < 15\%$). The errors generally increase with the increase of biogenic precursors in the mixture (with $|RE_i| < 30\%$ for the BIO mixture). In particular, for the summer LNOx scenarios, VBS-GECKO overestimates OA mass from the second day. This overestimation increases with the length of the simulation.

The origin of these errors is not entirely clear, as the role of many important driving variables is masked by the training and fitting procedures. It is however possible to list some of the major simplifications and approximations that were made in developing the VBS-GECKO parameterization: (i) discretization into 7 decadal bins (rather than continuous vapour pressure distribution for example); (ii) use of the same matrix of VB+OH stoichiometric coefficients to represent the reactivity of organic species formed from the oxidation of the precursor with OH, $O_3$ and $NO_3$; (iii) use of a single photolytic scale factor

(relative to acetone photolysis) to represent the overall photolysis of a complex mixture containing many photo-labile species; (iv) discretization into five α ratio values with separate optimization and linear interpolation to cover the entire range and; (v) selected simulation duration used for the optimization that is expected to influence the weight given at first or further generation products. More detailed consideration of these important parameters presents a clear opportunity to further improve the parameterization.





## 5 Conclusions

The GECKO-A modelling tool was used to explore SOA formation from the oxidation of various hydrocarbons in a box model under stationary state scenarios representative of environmental conditions. The set of parent hydrocarbons includes n-alkanes and 1-alkenes with 10, 14, 18, 22, and 26 carbon atoms, α-pinene, β-pinene and limonene, benzene, toluene, o-xylene, m-xylene and p-xylene. The developed environmental scenarios allow investigation of the sensitivity of SOA formation to changes of physical (photolysis, $C_{oa}$, T, relative humidity) and chemical (Ox/NOx/HOx) conditions. Simulated trends of maximum SOA yields are consistent with the literature data. In particular $Y_{max}$ increases with the carbon skeleton size of the parent the n-alkane or 1-alkenes, from 0.2 for the $C_{10}$ a plateau value of 1.2 for the $C_{>18}$. For the aromatics compounds, $Y_{max}$ slightly decrease with number of methyl groups with values below 0.15. For terpenes, simulated $Y_{max}$ increases in the following precursor's order: α-pinene, β-pinene and limonene. As expected, $Y_{max}$ increase when temperature decreases and $C_{OA}$ increases. NOx conditions drive the distributions of secondary organic species and changes substantially the volatility distribution of the species. Simulated SOA yields generally decreases with the increase of NOx (or α ratio). SOA yields from terpene are particularly sensitive to NOx levels, with values in the 1.1 - 1.75 range under low NOx condition (α close to 0) to values in the 0.05 – 0.2 range under high NOx conditions (α close to 1). SOA yields for species having a large carbon skeleton (i.e. with $n_C > 18$) show a low sensitivity to temperature, $C_{OA}$ and NOx regimes as they rapidly form compounds that partition dominantly to the condensed phase. Explicit simulations were used to explore the distribution of the millions of secondary organic species formed during the oxidation of the precursors in term of molar mass, vaporization enthalpy, OH reaction rate and Henry's law coefficient.

Simulation results obtained with the GECKO-A modeling tool were used as a reference to fit a VBS type parameterization. The VBS-GECKO parameterization is designed for 18 selected hydrocarbons and includes OH oxidation for all the hydrocarbons, as well as $NO_3$ and $O_3$ oxidation for the terpenes and 1-alkenes. For each hydrocarbon, 7 volatility bins are considered, ageing by reaction with OH, photolysis and partitioning according to the Raoult's law. NOx levels dependence of SOA formation is represented by a linear interpolation of different sets of parameters optimized for various α ratios. Properties of the bins, i.e. molar weight, vaporization enthalpy, OH rate constants and Henry's law coefficient are selected from the reference explicit simulations. The evaluation of the parameterization shows that VBS-GECKO captures the dynamic of SOA formation for a large range of conditions with a mean relative error on organic aerosol mass temporal evolution lesser than 20% compared to explicit simulations.

Explicit simulations of organic compound oxidation provided by GECKO-A have been previously used to develop a reduced chemical scheme for the HOx−NOx−VOC chemistry (Szopa et al., 2005) as well as to develop reduced parameterizations for SOA formation from semi-volatile and intermediate volatility n-alkanes (Hodzic et al., 2016). For the first time in this study, a reduction protocol has been developed and evaluated to optimize SOA parameterizations from explicit simulations for a large number of precursor species and under various environmental conditions. The optimization procedure has been automated to facilitate the update of the VBS-GECKO on the basis of the future GECKO-A versions, its extension to other



precursors and/or its modification to carry additional information (e.g. optical properties, hygroscopicity, N/C ratio). The implementation and the evaluation of VBS-GECKO in a 3D model is the purpose of a companion paper.

## Acknowledgements

This work was financially supported by the French Environment and Energy Management Agency (ADEME) and INERIS. We also thank French Ministry of Ecology for its financial support. This publication was developed under Assistance Agreement No.83587701-0 awarded by the U.S. Environmental Protection Agency to A. Hodzic. It has not been formally reviewed by EPA. The views expressed in this document are solely those of authors and do not necessarily reflect those of the Agency.

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



**Table 1: Constant environmental conditions used in the various scenarios. For each scenario, the concentrations of OH, HO$_2$, NO and NO$_2$ are at the stationary state induced by the constant conditions. Values in bold represent the conditions used in the "reference scenario".**

| Environmental parameter | | |
|---|---|---|
| Temperature | 270 - **298** | K |
| Relative humidity | **70** | % |
| Solar zenith angle | 30 - **50** - 70 | ° |
| C$_{oa}$ | 0,1 - **1** - 10 | µg m$^{-3}$ |
| [O$_3$] | 10 - **40** - 70 | ppb |
| [NO$_X$] | from 10$^{-2}$ to 10$^2$ (**1.25**) | ppb |
| [CO] | **120** | ppb |
| [CH$_4$] | **1750** | ppb |
| OH lifetime with the surrogate VOC | **1** | s |



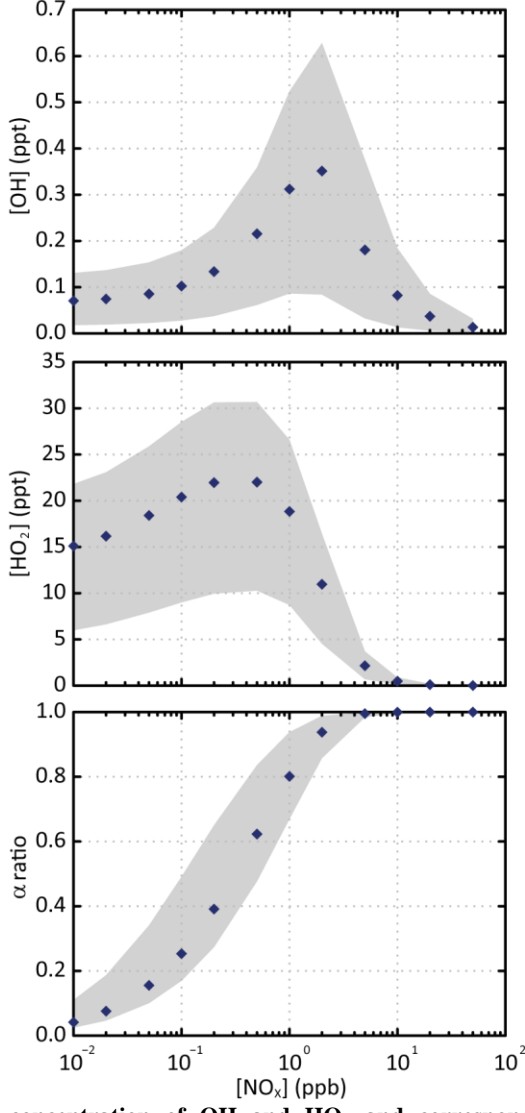

**Figure 1: Simulated stationary state concentration of OH and HO$_2$ and corresponding α values as a function of NOx concentrations for the various scenarios. The blue points are the simulated values for the scenario performed at a temperature of 298K, a zenith angle of 50° and 40 ppb of ozone. The grey zone corresponds to the range of values simulated in all the scenarios.**



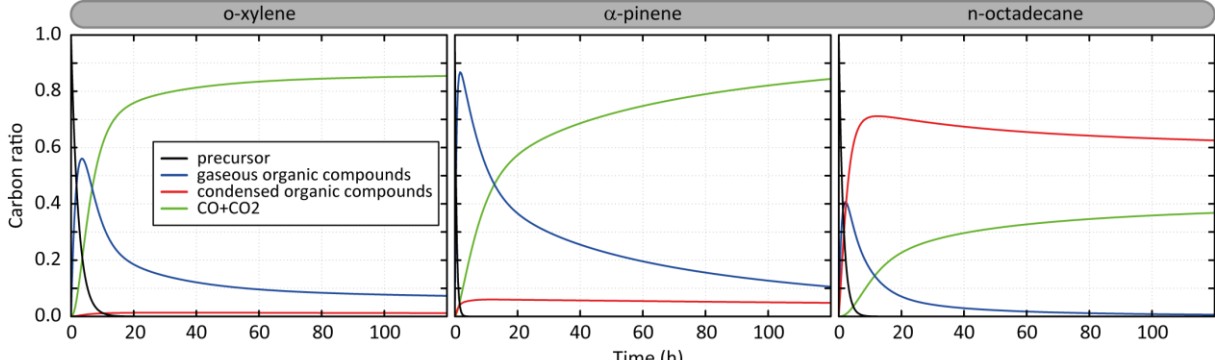

**Figure 2: Temporal evolution of the simulated carbon distribution between the precursor, CO and CO₂ and gaseous and condensed organic compounds. Simulated results are shown for o-xylene, α-pinene and n-octadecane for the scenario performed at a temperature of 298K, a zenith angle of 50°, a $C_{oa}$ of 1 μg m⁻³, 40 ppb of ozone and a α of 90%.**





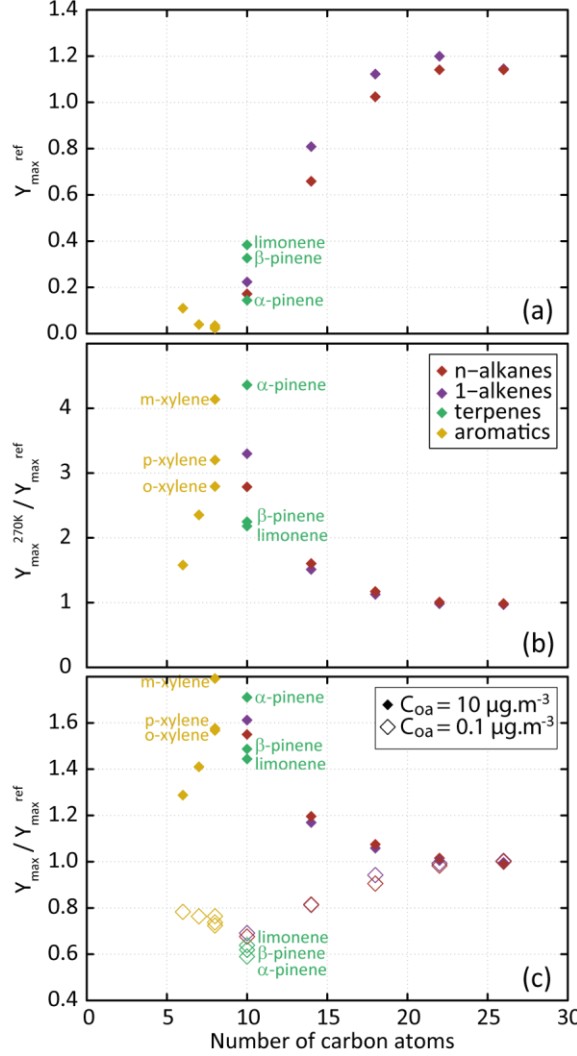

**Figure 3: (a) Dependence of $Y_{max}$ on the precursor's carbon number for the scenario performed at a temperature of 298K, a zenith angle of 50°, a $C_{oa}$ of 1 µg m$^{-3}$, 40 ppb of ozone and a α of 90%. (b) Dependence of $Y_{max}^{270K}/Y_{max}^{ref}$ on the precursor's carbon number. (c) Dependence of $Y_{max}^{0.1µgm-3}/Y_{max}^{ref}$ (open symbols) and $Y_{max}^{10µgm-3}/Y_{max}^{ref}$ (filled symbols) on the precursor carbon number.**



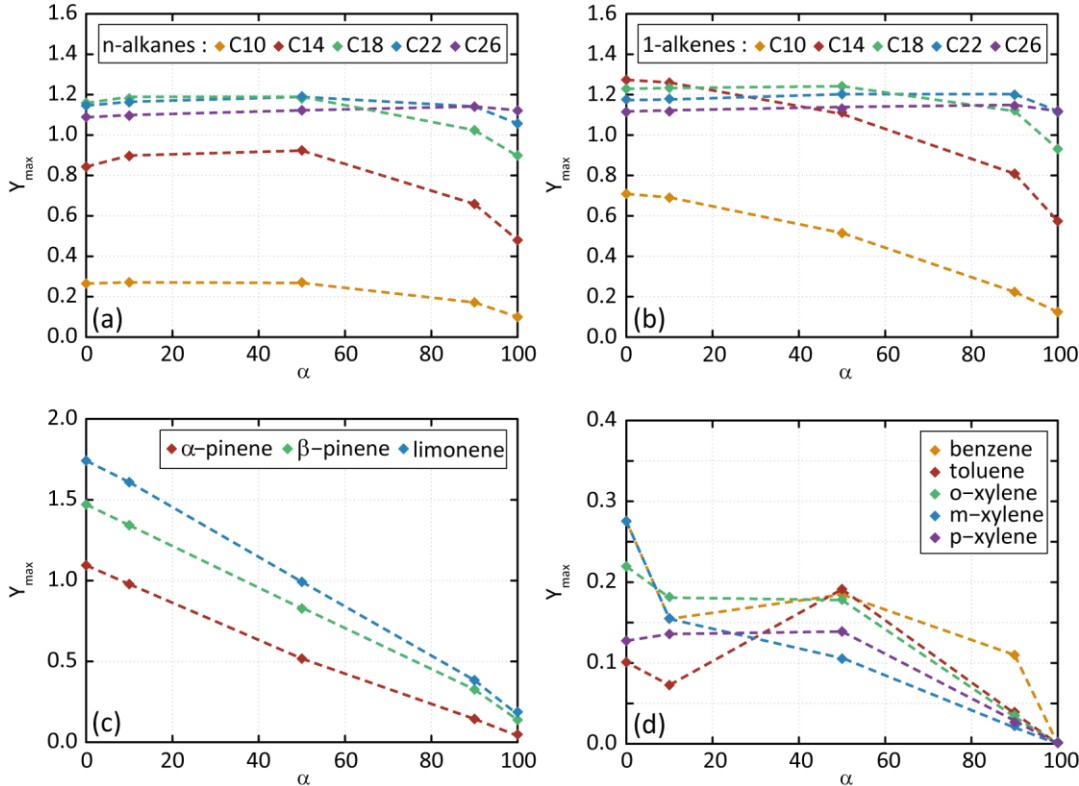

**Figure 4: Dependence of $Y_{max}$ on α for n-alkanes (a), 1-alkene (b), terpenes (c) and aromatic species (d) for the scenarios performed at a temperature of 298 K, a zenith angle of 50°, a $C_{oa}$ of 1 μg.m$^{-3}$ and 40 ppb of ozone. Points are connected with dashed lines to help in seeing trends of $Y_{max}$ with α.**





**Figure 5: Distribution of organic species (gaseous and particulate summed) in term of volatility ($P^{sat}_{298K}$) and (a) molar mass (Mw), (b) vaporisation enthalpy ($\Delta H_{vap}$), (c) OH reactivity ($k_{OH}$) and (d) solubility ($H^{eff}_{298K}$). Simulated results are shown for o-xylene, α-pinene and n-octadecane after 6h of oxidation for the scenario performed at a temperature of 298K, a zenith angle of 50°, a $C_{oa}$ of 1 µg m$^{-3}$, 40 ppb of ozone and a α of 90%. Each bubble is a species and the volume of the bubble is proportional to the carbon ratio represented by the species. The bubbles are coloured by volatility classes (see Table 2 for the bin bounds). The distribution of the**





properties weighted by the concentration in carbon atoms of the species in each volatility class at 6h is represented as box plots showing the 25, 50 and 75 percentiles and the minimal and maximal values. Red points are the mean properties weighted by the concentration in carbon atoms of the species in each volatility class.

5 **Table 2: Properties of the seven bins of the VBS-GECKO parameterization**

| | Bin bounds ($P^{sat}_{298K}$) | $P^{sat}_{298K}$ | Molar Weight | Vaporization Enthalpy | $H^{eff}_{298K}$ | $k_{OH}$ |
|---|---|---|---|---|---|---|
| | atm | atm | g mol$^{-1}$ | kJ mol$^{-1}$ | mol L$^{-1}$ atm$^{-1}$ | molec$^{-1}$ cm$^3$ s$^{-1}$ |
| VB1 | ]$10^{-7.5}$ ; $10^{-5.5}$] | $10^{-6.5}$ | 210 | 90 | $10^6$ | $4.10^{-11}$ |
| VB2 | ]$10^{-8.5}$ ; $10^{-7.5}$] | $10^{-8}$ | 240 | 105 | $10^7$ | $4.10^{-11}$ |
| VB3 | ]$10^{-9.5}$ ; $10^{-8.5}$] | $10^{-9}$ | 270 | 115 | $10^8$ | $4.10^{-11}$ |
| VB4 | ]$10^{-10.5}$ ; $10^{-9.5}$] | $10^{-10}$ | 300 | 125 | $10^9$ | $4.10^{-11}$ |
| VB5 | ]$10^{-11.5}$ ; $10^{-10.5}$] | $10^{-11}$ | 330 | 135 | $10^{10}$ | $4.10^{-11}$ |
| VB6 | ]$10^{-12.5}$ ; $10^{-11.5}$] | $10^{-12}$ | 360 | 145 | $10^{11}$ | $4.10^{-11}$ |
| VB7 | [$10^{-24}$ ; $10^{-12.5}$] | $10^{-14}$ | 390 | 165 | $10^{12}$ | 0 |

**Table 3: Simulation conditions of the learning scenarios**

| | Temperature | $C_{OA}$ | [OH] | [O$_3$] | [NO$_3$] | Reactivity of the precursor | Photolysis |
|---|---|---|---|---|---|---|---|
| | K | µg.m$^{-3}$ | ppt | ppb | ppt | | θ in ° |
| OH Chemistry | 270 | 0.1 | Induced by the conditions | 40.0 | Induced by the conditions | with OH only | No Photolysis reaction for SOCs θ = 50° |
| | | 1.0 | | | | | |
| | | 10.0 | | | | | |
| | 298 | 0.1 | | | | | |
| | | 1.0 | | | | | |
| | | 10.0 | | | | | |
| O$_3$ Chemistry | 298 | 1.0 | Induced by the conditions | 10.0 | Induced by the conditions | with O$_3$ only | No Photolysis reaction for SOCs θ = 50° |
| | | | | 40.0 | | | |
| | | | | 70.0 | | | |
| NO$_3$ Chemistry | 298 | 1.0 | Induced by the conditions | 40.0 | 0.04 (only for terpenes) | with NO$_3$ only | No Photolysis reaction for SOCs θ = 50° |
| | | | | | 0.4 | | |
| | | | | | 40.0 (only for n-alkenes) | | |
| Photolysis | 298 | 1.0 | Induced by the conditions | 40.0 | Induced by the conditions | with OH, O$_3$ and NO$_3$ | θ = 30° |
| | | | | | | | θ = 50° |
| | | | | | | | θ = 70° |





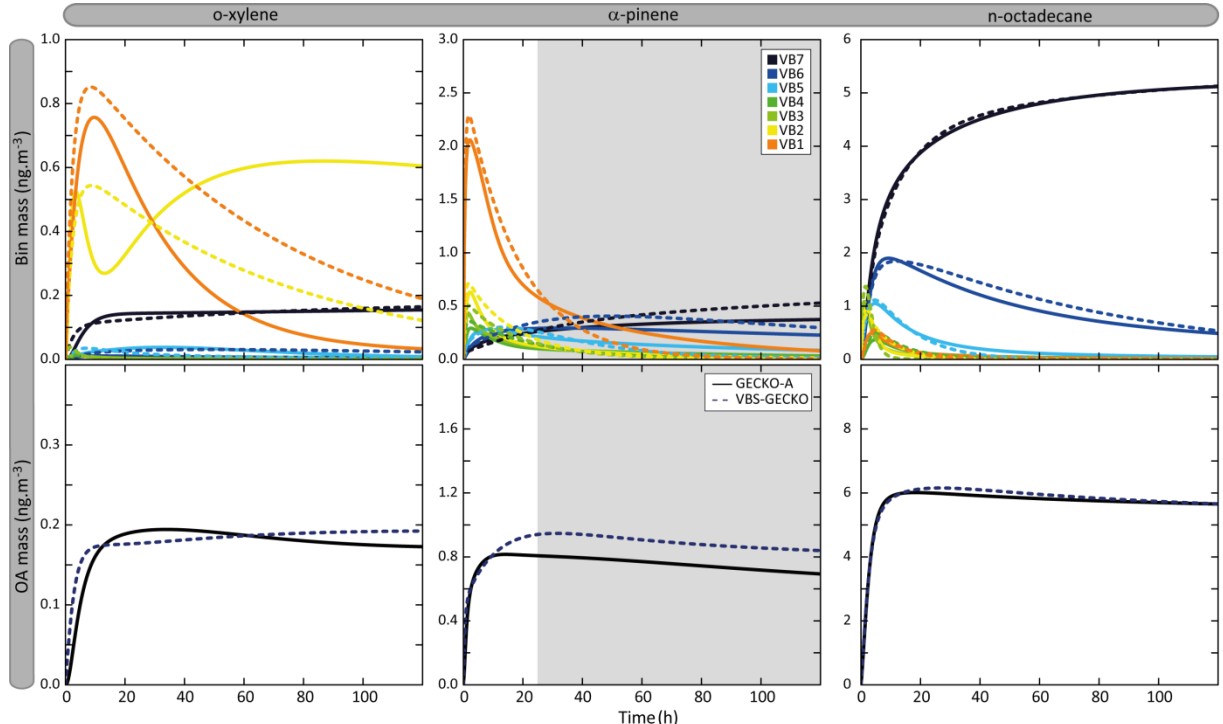

**Figure 6: Temporal evolution of volatility bin mass concentration (top panels) and organic aerosol mass (bottom panel) simulated with GECKO-A (continuous lines) and VBS-GECKO (dashed lines). Simulated results are shown for o-xylene, α-pinene and n-octadecane for the scenario performed at a temperature of 298 K, a zenith angle of 50°, a $C_{oa}$ of 1 µg.m$^{-3}$, 40 ppb of ozone and a α of 90%. Parameters have been optimized on duration represented by white background on the figure.**




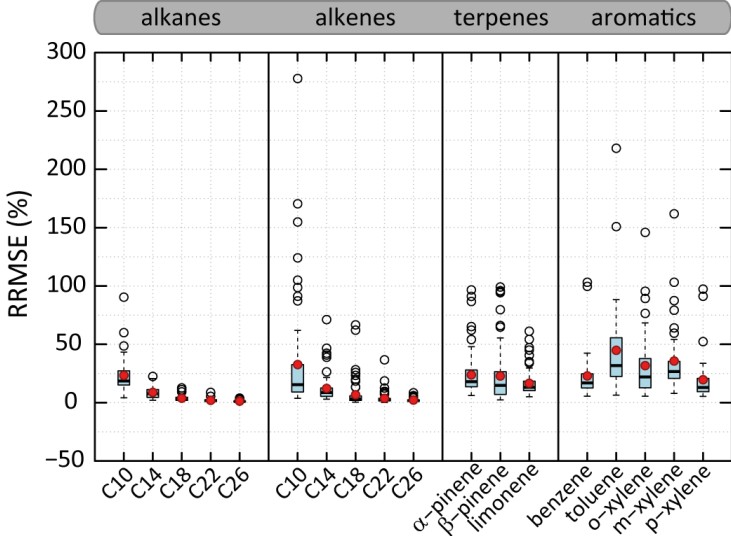

**Figure 7: Distributions of the RRMSE on organic aerosol mass concentration calculated between VBS-GECKO and GECKO-A on the learning simulations. The bottom and top of the boxes are the first and third quartiles, the bands inside the boxes represent the medians. The ends of the whiskers represent the lowest RRMSE still within 1.5 × inter-quartile range (IQR) of the lower quartiles, and the highest RRMSE still within 1.5 IQR of the upper quartiles. Red points represent the mean values and empty points the outliers. The number of data is of 45 simulations for n-alkane or aromatic and 70 for 1-alkene or terpene. Time steps for which the OA mass simulated with GECKO-A is under a threshold of $5 \times 10^{-5}$ µg.m-3 (i.e. a SOA yield lower than around 1%) are not considered in RRMSE calculation.**




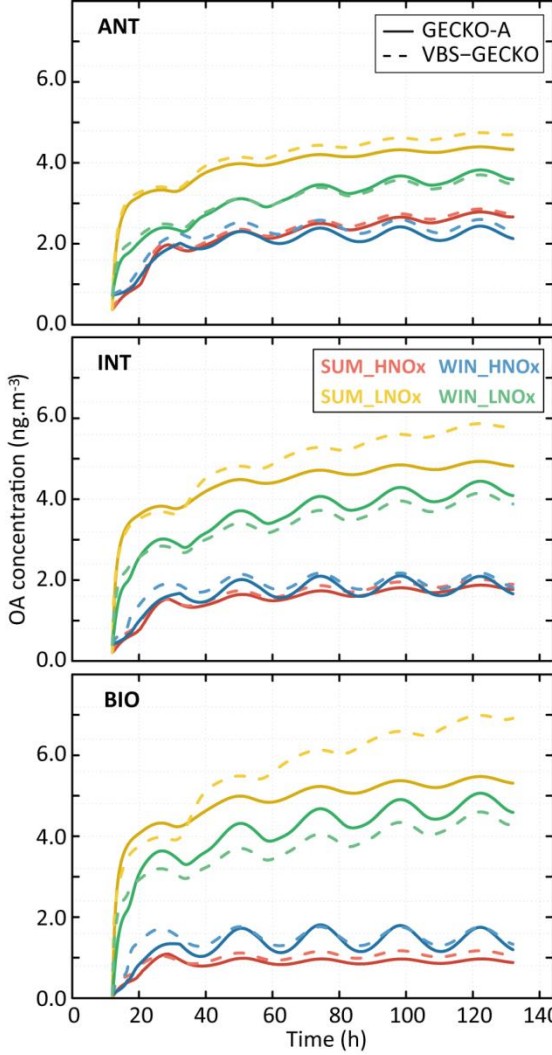

**Figure 8: Simulated organic aerosol (OA) mass with GECKO-A (continuous lines) and VBS-GECKO (dashed lines), for the different evaluation scenarios using $C_{oa}$ = 2 µg.m$^{-3}$. Top panel presents results for the ANT cases, middle panel for the INT cases and bottom panel for the BIO cases. The different colours represent the different scenarios (see text).**




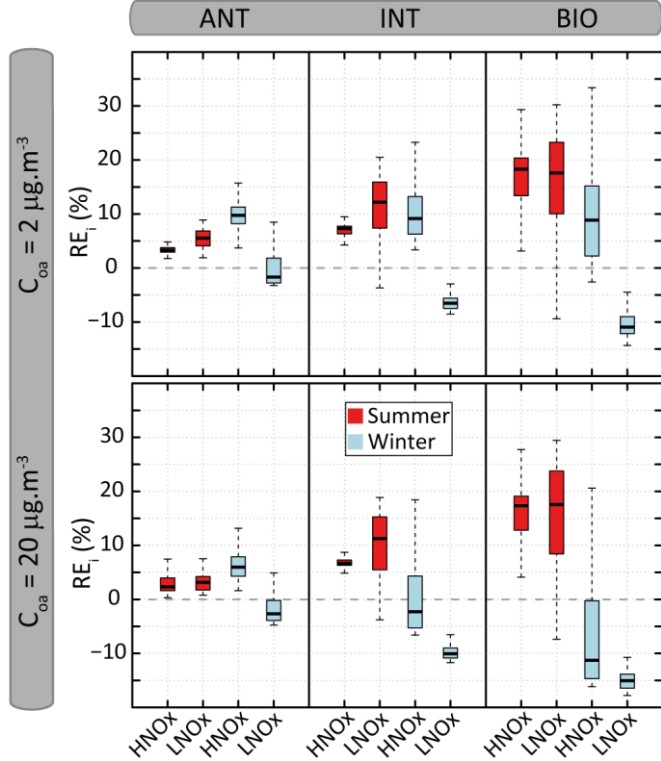

**Figure 9 - Distribution of the relative errors (RE$_i$) on organic aerosol mass concentration calculated between VBS-GECKO and GECKO-A for the 24 evaluation scenarios. The bottom and top of the boxes are the first and third quartiles, the bands inside the boxes represent the medians. The ends of the whiskers represent the lowest and the highest RE$_i$. The number of data is of 361 time steps for each evaluation scenario.**