# Peer review of "Exploration of the influence of environmental conditions on secondary organic aerosol formation and organic species properties using explicit simulations: development of the VBS-GECKO parameterization"

_Atmospheric Chemistry and Physics, 2018_

## Short Comment (SC1) · 16 May 2018

Estimation of ODE parameters for a system of equations is a challenging problem. Can you please provide details as to how the BObyQA method was applied for this particular problem? Were any idealized tests (i.e. solving the ode's with known parameters and then using the numerical solutions to retrieve the parameters) performed for typical conditions to test the robustness and accuracy of the fitting approach in estimating

parameters of the ODEs?

Are the learning scenario data publicly available?

---

## Referee Comment (RC1) · Anonymous Referee #1 · 24 May 2018

This work develops a simplified VBS-GEcko parameterization determined by fitting a lumped set of volatility bins to the explicit Gecko_A framework. Optimizations are performed by defining design test criteria in terms of VOC precursor, NOx, photolysis etc. The ultimate objective is to implement VBS-Geck in 3D chemical transport models. Overall, the paper is well written.

While the goals and methodologies are well defined, the following points need to be

explained: 1. While VBS-Gecko is compared to Gecko_A, it's not clear how either of these compares to actual SOA observations. Simulating some lagrangian test cases from field measurements of SOA would have been helpful to understand the utility of these approaches. 2. A fixed set of yields are fit to different volatility bins in VBS-Gecko. But multigenerational aging e.g. functionalization/fragmentation reactions can change the VBS distributions, especially at longer timescales. Could use of fixed VBS-Gecko (which do not change with aging) yields be responsible for the differences with explicit Gecko-A shown in Figure 8? I understand that the fixed yields are supposed to represent a fit to the entire dynamic evolution, but errors could be larger at longer timescales. 3. Condensed phase SOA processes as oligomerization can also alter volatility distributions, molar mass etc. How does VBS-Gecko or Gecko-A account for dynamic changes in SOA properties due to condensed phase chemistry? 4. In Figure 2, why is the condensed mass of n-octadecane (red line) much higher than the other 2 precursors? Is this result supported by smog chamber measurements? 5. Mechanistically, why does maximum yield decrease as the number of methyl groups in aromatics increase? 6. The enthalpies of vaporization are assumed to be NOx-independent and only depend on volatility bins. However, volatility of SOA is NOx-dependent. See Xu et al. 2014. Could the authors comment on how NOx-dependent volatility affects their assumptions of SOA properties e.g. enthalpies of vaporizations and molar mass? 7. Why do terpene SOA yields show the strongest sensitivity to temperature and pre-existing organic aerosol mass? 8. I recommend adding references to some recent review papers in the context of challenges in SOA measurements and modeling in the Introduction e.g. (Ng et al. 2017, Shrivastava et al. 2017).

References:

Xu, L., Kollman, M. S., Song, C., Shilling, J. E. & Ng, N. L. Effects of NOx on the Volatility of Secondary Organic Aerosol from Isoprene Photooxidation. Environ. Sci. Technol. 48, 2253-2262, doi:10.1021/es404842g (2014).

Ng, N. L. et al. Nitrate radicals and biogenic volatile organic compounds: oxidation, mechanisms, and organic aerosol. Atmos. Chem. Phys. 17, 2103-2162, doi:10.5194/acp-17-2103-2017 (2017).

Shrivastava, M. et al. Recent advances in understanding secondary organic aerosol: Implications for global climate forcing. Rev. Geophys. 55, 509-559, doi:10.1002/2016RG000540 (2017).

---

## Referee Comment (RC2) · Anonymous Referee #2 · 15 Jun 2018

The Lannuque et al. manuscript reports on the use of the GECKO-A model (Generator of Explicit Chemistry and Kinetics of Organics in the Atmosphere) to develop Volatility Basis Set (VBS) parameterizations of secondary organic aerosol (SOA) formation for use in three-dimensional chemical transport models. Initial chemical conditions for the SOA simulations were generated by running GEKCO-A using fixed concentrations of CO and methane, and a range of NOx and O3 values. For a given O3 value, each NOx value leads to a steady-state OH and HO2 value, which are then used as

the chemical inputs for the SOA parameter optimization simulations (along with the CO, methane, NOx, and O3 values). Simulations were also conducted to consider a range of temperatures and photolysis rates (set by the solar zenith angle), as well as a range of pre-existing organic aerosol mass concentrations. The simulation results were used to then optimize the number and properties of the VBS bins, as well as the stoichiometric coefficients (leading to mass contributions) in each bin. A total of 7 bins were selected; the stoichiometric coefficients/concentration in each bin is a function of precursor and reaction conditions. The GECKO-VBS parameterization was evaluated against GECKO-A using a test set of simulations (independent of the optimization set). The GECKO-VBS parameterization represents general trends in SOA formation, and performs best in the majority anthropogenic precursor simulations; SOA is over-predicted in the majority biogenic precursor simulation with low NOx. The process for developing VBS parameterizations using GECKO-A was automated and should be of significant benefit to the community in the future.

In summary, GECKO-A is a unique modeling tool in its near-explicit representation of gas-phase chemistry. A complementary approach to experimentally-based methods for developing parameterizations of SOA is presented in this work. Some combination of GECKO-A type modeling as presented and experimental studies likely will lead to the best SOA parameterizations for three-dimensional model predictions. GECKO-A/-VBS model simulations additionally provide a unique metric against which to compare more chemically explicit gas- and particle-phase compositional data. This manuscript should be of great interest to the ACP readership. It is generally well written and easy to follow. Specific technical and editorial comments follow.

Technical: In regard to the initial condition simulations, it is stated that the NOx and HOx concentrations are typical of chemical characteristics of low- to high-NOx environments (p. 4, l. 16) and it was found that for NO levels > 1 ppb most of the RO2 reacts with NOx. This observed branching ratio is different than has been reported for regional (Barsanti et al., ACP, 2013, doi:10.5194/acp-13-12073-2013) and global

(Henze et al., ACP, 2008, https://www.atmos-chem-phys.net/8/2405/2008/) model simulations. This point is worth further discussion in the manuscript. Are the NOx and HOx concentrations truly representative of the ambient atmosphere? Under what limitations/conditions? What are the reasons that the GECKO-A model simulations of this branching ratio differ from those generated using a condensed gas-phase chemical mechanism? Is one more representative than the other? What are the implications for predicted SOA formation?

In the description of the treatment of partitioning (p. 5, l. 14-24), it is stated that 250 g mol-1 is used as the mean MW of the condensed phase. Is this for all of the GECKO simulations (e.g., GECKO-A and GECKO-VBS)? If so, why? Given that mean MW can be explicitly calculated and the MW distribution shown in Fig. 5 and related discussion (e.g., p. 9, l. 30) suggests a higher value is supported by the model simulations. What is the role of RH in the simulations? Does RH affect the partitioning constants (e.g., modify mean MW)? Or does it affect deposition?

In defining the seven VBS bins, the same properties are assigned to each bin, regardless of the precursor. In the case of the kOH values, as discussed in the manuscript, this is justifiable and unlikely to significantly affect the parameterizations or simulated SOA concentrations. However, the assignment of the Henry's Law Constants needs further discussion and justification. In Table 2, the effective Henry's Law Constants increase for each bin as volatility decreases. For the simulation results shown in Fig. 5, this trend only seems to hold for alpha-pinene (looking at the bin mean); further, between the precursors, the Henry's Law Constants (again, bin mean) vary by orders of magnitude between the precursors for the same volatility bin. Given that Henry's Law Constants are often used in wet and dry deposition parameterizations, this approach needs further consideration. In addition, in regard to the GECKO-A/GECKO-VBS comparisons, depending on the importance of dry deposition in the simulations, this may explain some of the disagreement.

On p. 7, l. 31-32 it is stated that the species formed at high NOx have a molar mass

that is $\sim$ 100 g mol-1 higher than at low NOx. Was this value calculated across all bins and all precursors? A visual inspection of Fig. S1 does not necessarily support this statement. Further, even if numerically true, it seems an unnecessary oversimplification (particularly since the Figure is in the supplement). The changes in molar mass seem to vary significantly between alpha values and across bins/precursors.

Editorial: It is recommend that the symbol beta be used to describe the reaction of RO2 with NO relative to HO2, as in Pye et al. (ACP, 2010, doi:10.5194/acp-10-11261-2010). The symbol alpha is confusing, particularly in the Supplementary Table S1, since alpha in SOA parameterizations is historically used to represent the stoichiometric coefficients.

It is also recommended that since the abbreviation OA for organic aerosol is introduced, it should be used throughout the manuscript (see for example lines 26-31 on p. 12).

Abstract-line 29: "dynamic" should be "dynamics". The discussion of the VBS-GEKCO performance is awkward as written. Maybe just, "In evaluating the ability of VBS-GECKO to capture the temporal evolution of SOA mass, the mean relative error is less than 20% compared to GEKCO-A."

p.2, line 16: "Theirs" should be "their". What does "their" refer to? This is awkward as written (description of SOA parameterization). Line 19: I don't think that Cappa and Wilson 2012 use decadal volatility bins. This should be checked. Line 30: "influences" should be "influence"

p. 6, line 16: "growths" should be "increases"

p. 8, line 22: "redistributing" should be "redistributes"

p. 11, line 3: "performed on" should be "performed for" p. 12, line 19: It is suggested that "do however not" be reworded as "do not, however, partition"; line 31: "lesser" should be "less"

p. 13, line 19: add "respectively" after scenario; line 31: It is suggested that "species

was arbitrary" be replaced with "species was arbitrarily set to the same"

p. 15, line 9: "aromatics compounds" should be "aromatic compounds"; line 22-23: It is suggested that "NOx levels dependence of SOA formation" be replaced with "The dependence of SOA formation on NOx levels is represented..."

---

## Author Comment (AC1) · 18 Jul 2018

We thank the reviewers for their comments on the manuscript. We outline below responses to the points raised by each referee and summarize the changes made to the revised manuscript.

RESPONSES TO REFEREE 1

[Figure]

<1. While VBS-Gecko is compared to Gecko_A, it's not clear how either of these compares to actual SOA observations. Simulating some lagrangian test cases from field measurements of SOA would have been helpful to understand the utility of these approaches.>

The paper focuses on the development of the VBS-GECKO only. However, some observation/simulation comparisons for a few plume conditions were already performed in the past using the GECKO-A modeling tool (e.g. Lee-Taylor et al., 2011, 2015; Hodzic et al., 2013). Furthermore, the VBS-GECKO has been implemented in the chemistry-transport model CHIMERE and results compared with both in-situ measurements and other model outputs. Results show that the use of the VBS-GECKO parameterization slightly improves the performances of the model on OA concentrations compared to the current parameterization in CHIMERE. Detailed results will be included in a companion paper, to be submitted in ACPD (Lannuque et al., 2018).

<2. A fixed set of yields are fit to different volatility bins in VBS-Gecko. But multi-generational aging e.g. functionalization/fragmentation reactions can change the VBS distributions, especially at longer timescales. Could use of fixed VBS-Gecko (which do not change with aging) yields be responsible for the differences with explicit Gecko-A shown in Figure 8? I understand that the fixed yields are supposed to represent a fit to the entire dynamic evolution, but errors could be larger at longer timescales.>

The sets of VBS-GECKO stoichiometric coefficients are optimized to reproduce both SOA formation and aging. As described in the paper (section 4.1), aging is described by gas phase reactions of the VBS-GECKO bins (VBx). All the coefficients of the precursor + oxidants and VBx+OH are fitted on simulations lasting 1, 2 or 5 days (depending on the lifetime of the precursor), based on GECKO-A generated chemical schemes involving 15 successive generations of non-radical compounds. VBS-GECKO should therefore represent the gas phase multigenerational aging on a time scale similar to SOA lifetime.

[Figure]

<3. Condensed phase SOA processes as oligomerization can also alter volatility distributions, molar mass etc. How does VBS-Gecko or Gecko-A account for dynamic changes in SOA properties due to condensed phase chemistry?>

The VBS-GECKO architecture depends directly on GECKO-A. Current version of the GECKO-A modeling tool ignores the aging process in the condensed phase, as stated in section 2.2 of the paper. Adding these processes explicitly require mechanistic and kinetic knowledge that still remain sparse. This is currently one of the main limits of the VBS-GECKO approach. Note that an empirically based parameterization could still later be added in VBS-GECKO to represent aging in the condense phase.

<4. In Figure 2, why is the condensed mass of n-octadecane (red line) much higher than the other 2 precursors? Is this result supported by smog chamber measurements?>

Octadecane have 10 and 8 carbon atoms more than o-xylene and $\alpha$-pinene, respectively. The vapor pressure of octadecane oxidation products is therefore typically lower than products from o-xylene and $\alpha$-pinene. Furthermore, less oxidation steps are required to form species of enough low volatility to condense, decreasing the formation of fragmented compounds (e.g. CO and CO2). These results are in agreement with smog chamber studies showing a much higher yields for long chain alkanes than for mono-aromatics and terpenes (e.g. Lee et al., 2006; Lim and Ziemann, 2009; Li et al., 2016; Ng et al., 2007).

<5. Mechanistically, why does maximum yield decrease as the number of methyl groups in aromatics increase?>

For compounds having an identical number of carbon atoms, increasing the number of methyl groups is known to promote fragmentation (e.g. Aumont et al., 2013) and therefore to decrease SOA yields. However, increasing the size of the carbon skeleton also leads to lower volatility species and thus increases SOA yields (e.g. Lim and Ziemann, 2009). For the aromatic series, the increase of fragmentation outweighs the

decrease of volatility of the parent compound. This trend is consistent with chamber observations (e.g. Li et al., 2016).

<6. The enthalpies of vaporization are assumed to be NOx-independent and only depend on volatility bins. However, volatility of SOA is NOx-dependent. See Xu et al. 2014. Could the authors comment on how NOx-dependent volatility affects their assumptions of SOA properties e.g. enthalpies of vaporizations and molar mass?>

In VBS-GECKO, SOA volatility is NOx-dependent owing to the different sets of stoichiometric coefficients for various levels of NOx (see section 4.3.1). Indeed, the relative contribution of different volatility bins in SOA production changes according to the NOx conditions. The estimation method of Nannoolal et al. (2008) is used to calculate both the enthalpies of vaporization and the vapor pressure of the species. This method leads to a strong correlation between the two parameters (see Fig. 5) and justify the use of a mean vaporization enthalpy per volatility bin. Moreover, molar masses and enthalpies of vaporization have been fixed before fitting the coefficients to allow compensation of possible biases. Results of the fits at 298 and 270 K show that the approach captures well the sensitivity of SOA to temperature. Finally, even though vaporization enthalpies are fixed, the overall SOA volatility (and their sensitivity to temperature changes via the prescribed vaporization enthalpies) depends on the concentration of each bin, which is NOx dependent.

<7. Why do terpene SOA yields show the strongest sensitivity to temperature and pre-existing organic aerosol mass?>

Figure 3 shows that a high sensitivity of SOA yields to temperature and pre-existing organic aerosol mass is a common feature for precursors having a skeleton with 8-12 carbon atoms. For these precursors, SOA contributors fall mostly in the VB4 and VB5 bins, i.e. with a saturation vapor pressure in the 109-1011 atm range. Gas/particles partitioning of these bins are the most sensitive to temperature and pre-existing organic aerosol mass.

<8. I recommend adding references to some recent review papers in the context of challenges in SOA measurements and modeling in the Introduction e.g. (Ng et al. 2017, Shrivastava et al. 2017).>

The two references were added.

RESPONSES TO REFEREE 2

<In regard to the initial condition simulations, it is stated that the NOx and HOx concentrations are typical of chemical characteristics of low- to high-NOx environments (p. 4, l. 16) and it was found that for NO levels > 1 ppb most of the RO2 reacts with NOx. This observed branching ratio is different than has been reported for regional (Barsanti et al., ACP, 2013, doi:10.5194/acp-13-12073-2013) and global (Henze et al., ACP, 2008, https://www.atmos-chem-phys.net/8/2405/2008/) model simulations. This point is worth further discussion in the manuscript. Are the NOx and HOx concentrations truly representative of the ambient atmosphere? Under what limitations/conditions? What are the reasons that the GECKO-A model simulations of this branching ratio differ from those generated using a condensed gas-phase chemical mechanism? Is one more representative than the other? What are the implications for predicted SOA formation?>

We do not fully agree that our simulated levels of HO2, RO2 for the various prescribed NOx concentrations (and therefore RO2+NO vs RO2+HO2 reaction rates) strongly differ from other atmospheric simulations results. The various models typically use a similar NOx/HOx/NOx/CO/CH4 chemistry, which is also used to define our modeling scenario (see section 2.1). Depending on the scenarios (from remote to urban), an additional simplified organic "module" is used as a proxy for the key contribution of VOC in the chemistry of the HOx pool. Trend and magnitude of the simulated HO2 versus NOx concentration (see figure 1) appears to be consistent with observations, e.g. Stone et al. (2012) (see Fig. 9 of Stone et al., 2012). Furthermore, the calculated reaction rate ratio $\alpha$ for the various NO concentrations (now denoted RRR, see below) agree fairly

[Figure]

well with those presented by Henze et al., 2008 (see for example the Fig. 4 of Henze et al., 2008).

Two levels of NOx are considered in the parameterization of Henze et al. (2008) and Barsanti et al. (2013): either a high or a low NOx case. In the VBS-GECKO parameterization, 5 distinct levels of NOx are considered, selected to cover the full range of $\alpha$ (or RRR) values (see section 4.3.1) and accounting for the nonlinear relationship between these 2 parameters. Our approach also avoid the issues of associating "experimental low/high-NOx conditions" with "atmospheric low/high-NOx conditions", as presented by Barsanti et al. (2013) in their article.

<In the description of the treatment of partitioning (p. 5, l. 14-24), it is stated that 250 g mol-1 is used as the mean MW of the condensed phase. Is this for all of the GECKO simulations (e.g., GECKO-A and GECKO-VBS)? If so, why? Given that mean MW can be explicitly calculated and the MW distribution shown in Fig. 5 and related discussion (e.g., p. 9, l. 30) suggests a higher value is supported by the model simulations. What is the role of RH in the simulations? Does RH affect the partitioning constants (e.g., modify mean MW)? Or does it affect deposition?>

A mean molecular weight of 250 g mol-1 was only set to the seed (i.e. preexisting) OA to compute the gas/particle partitioning of the species according to the Raoult's law, as described page 5 of the paper. This value is used in all the simulations performed in this study. The pre-existent OA is not further taken in account in the analysis (as in Fig. 5) and to set the properties of the various bins (VBx). Furthermore, deposition is ignored for organic matter (gas and particles). Mean Henry's law coefficient Heff are computed for the various bin (see table 2) for the later application of the parameterization in 3D models (where Heff is usually a required parameter in deposition module). In the current version of the model, effect of Relative Humidity RH on the partitioning of organic species is ignored. In the box model, RH only contributes to the HOx chemistry (in particular for the production of OH from O3 photolysis)

<In defining the seven VBS bins, the same properties are assigned to each bin, regardless of the precursor. In the case of the kOH values, as discussed in the manuscript, this is justifiable and unlikely to significantly affect the parameterizations or simulated SOA concentrations. However, the assignment of the Henry's Law Constants needs further discussion and justification. In Table 2, the effective Henry's Law Constants increase for each bin as volatility decreases. For the simulation results shown in Fig. 5, this trend only seems to hold for alpha-pinene (looking at the bin mean); further, between the precursors, the Henry's Law Constants (again, bin mean) vary by orders of magnitude between the precursors for the same volatility bin. Given that Henry's Law Constants are often used in wet and dry deposition parameterizations, this approach needs further consideration. In addition, in regard to the GECKO-A/GECKO-VBS comparisons, depending on the importance of dry deposition in the simulations, this may explain some of the disagreement.>

Indeed, the distribution of Henry's coefficient within each bin spans several orders of magnitude. Representing this additional information would require the development of 2 dimensional VBS, to capture both the volatility and solubility distributions. This is out of the scope of the present parameterization. However, a clear trend is observed for the mean Heff, increasing as volatility decrease. This trend is not specific to a-pinene but is also found for the other precursors (note that results shown in Fig. 5 only represent the distributions of properties at a given time for 3 simulations among 1010 simulations lasting five days). The mean Heff was computed for each bin, based on all training simulations performed for all the alkane parent compounds (see page 9). As stated above, Heff values are not used for the box model simulations performed in this study but provided for the application of VBS-GECKO in 3D chemical transport model. In a second article in preparation (Lannuque et al., 2018), VBS-GECKO is implemented in a chemical-transport model and the sensitivity of simulated SOA concentrations to Henry's law Constants is examined (including effect on both partitioning and deposition). First results show a weak sensitivity on simulated SOA concentration to variation of Heff.

[Figure]

<On p. 7, l. 31-32 it is stated that the species formed at high NOx have a molar mass that is ≈ 100 g mol-1 higher than at low NOx. Was this value calculated across all bins and all precursors? A visual inspection of Fig. S1 does not necessarily support this statement. Further, even if numerically true, it seems an unnecessary oversimplification (particularly since the Figure is in the supplement). The changes in molar mass seem to vary significantly between alpha values and across bins/precursors.>

The statement concerning the "100 g mol-1" has been removed. Indeed, that statement is more qualitative than quantitative.

<Editorial: It is recommend that the symbol beta be used to describe the reaction of RO2 with NO relative to HO2, as in Pye et al. (ACP, 2010, doi:10.5194/acp-10-11261-2010). The symbol alpha is confusing, particularly in the Supplementary Table S1, since alpha in SOA parameterizations is historically used to represent the stoichiometric coefficients.>

The term "$\alpha$ ratio" has been replaced by "RRR" for "reaction rate ratio" of RO2 to remove all ambiguity.

<p.2, line 16: "Theirs" should be "their". What does "their" refer to? This is awkward as written (description of SOA parameterization).>

"Their" refers to "secondary organic species".

<Line 19: I don't think that Cappa and Wilson 2012 use decadal volatility bins. This should be checked.>

Yes, Cappa and Wilson 2012 don't use decadal volatility bins (a miss placed reference) the references was removed.

The other editorial recommendations were taken into account in the manuscript. Thanks for the suggested corrections.

OTHER COMMENTS:

<Estimation of ODE parameters for a system of equations is a challenging problem. Can you please provide details as to how the BObyQA method was applied for this particular problem? Were any idealized tests (i.e. solving the ode's with known parameters and then using the numerical solutions to retrieve the parameters) performed for typical conditions to test the robustness and accuracy of the fitting approach in estimating parameters of the ODEs?>

The ODE solving method applied is the "lsoda" function provided in the "deSolve" R package. The BObyQA optimization method applied in this study is a non-modified "BObyQA" function provided in the "nloptr" R package. The BObyQA method offers the possibility to constrain the coefficients (here bounds are set to 0-1 for the stoichiometric coefficients and 0-100 for the photolysis factors, see section 4.3.1). No idealized test has been performed. Nevertheless, many different optimization methods have been tested on simpler cases before selecting the method. These tests were performed on single simulation, using a C14 alkane as the parent compound, without temperature, NOx or Coa variations. Various configuration of VBS-GECKO were then tested, including different number of bins. The BObyQA was found to be especially robust, providing the same set of optimized coefficients for different sets of initial coefficients used to start the iteration process. The consistency of the set of ODE and its ability to capture SOA formation was systematically examined during the progressive design of the VBS-GECKOA parameterization.

<Are the learning scenario data publicly available?>

The learning scenario data involve too many species and conditions ($\sim 6.5 \times 10^{10}$ values) and have not been formatted to be published online. These dataset are available on request.

REFERENCES:

Aumont, B., Camredon, M., Mouchel-Vallon, C., La, S., Ouzebidour, F., Valorso, R., Lee-Taylor, J. and Madronich, S.: Modeling the influence of alkane molecu-

lar structure on secondary organic aerosol formation, Faraday Discuss., 165, 105, doi:10.1039/c3fd00029j, 2013.

Barsanti, K. C., Carlton, A. G. and Chung, S. H.: Analyzing experimental data and model parameters: Implications for predictions of SOA using chemical transport models, Atmos. Chem. Phys., 13(23), 12073–12088, doi:10.5194/acp-13-12073-2013, 2013.

Cappa, C. D. and Wilson, K. R.: Multi-generation gas-phase oxidation, equilibrium partitioning, and the formation and evolution of secondary organic aerosol, Atmos. Chem. Phys., 12, 9505–9528, doi:10.5194/acp-12-9505-2012, 2012.

Henze, D. K., Seinfeld, J. H., Ng, N. L., Kroll, J. H., Fu, T.-M., Jacob, D. J. and Heald, C. L.: Global modeling of secondary organic aerosol formation from aromatic hydrocarbons: high- vs. low-yield pathways, Atmos. Chem. Phys., 8(9), 2405–2420, doi:10.5194/acp-8-2405-2008, 2008.

Hodzic, A., Aumont, B., Knote, C., Lee-Taylor, J., Madronich, S. and Tyndall, G.: Volatility dependence of Henry's law constants of condensable organics: Application to estimate depositional loss of secondary organic aerosols, Geophys. Res. Lett., 41(13), 4795–4804, doi:10.1002/2014GL060649, 2014.

Hodzic, A., Madronich, S., Aumont, B., Lee-Taylor, J., Karl, T., Camredon, M. and Mouchel-Vallon, C.: Limited influence of dry deposition of semivolatile organic vapors on secondary organic aerosol formation in the urban plume, Geophys. Res. Lett., 40(12), 3302–3307, doi:10.1002/grl.50611, 2013.

Lannuque, V., Couvidat, F., Camredon, M., Aumont, B. and Bessagnet, B.: Modelling of organic aerosol over Europe in summer conditions with the VBS-GECKO parameterization: sensitivity to secondary organic compound properties and IVOC emissions, Atmos. Chem. Phys. Diss., to be submitted, 2018.

Lee, A., Goldstein, A. H., Kroll, J. H., Ng, N. L., Varutbangkul, V., Flagan, R. C.

and Seinfeld, J. H.: Gas-phase products and secondary aerosol yields from the photooxidation of 16 different terpenes, J. Geophys. Res. Atmos., 111(17), 1–25, doi:10.1029/2006JD007050, 2006.

Lee-Taylor, J., Madronich, S., Aumont, B., Baker, A., Camredon, M., Hodzic, A., Tyndall, G. S., Apel, E. and Zaveri, R. A.: Explicit modeling of organic chemistry and secondary organic aerosol partitioning for Mexico City and its outflow plume, Atmos. Chem. Phys., 11(24), 13219–13241, doi:10.5194/acp-11-13219-2011, 2011.

Lee-Taylor, J., Hodzic, A., Madronich, S., Aumont, B., Camredon, M. and Valorso, R.: Multiday production of condensing organic aerosol mass in urban and forest outflow, Atmos. Chem. Phys., 15(2), 595–615, doi:10.5194/acp-15-595-2015, 2015.

Li, L., Tang, P., Nakao, S., Chen, C. L. and Cocker, D. R.: Role of methyl group number on SOA formation from monocyclic aromatic hydrocarbons photooxidation under low-NOx conditions, Atmos. Chem. Phys., 16(4), 2255–2272, doi:10.5194/acp-16-2255-2016, 2016.

Lim, Y. B. and Ziemann, P. J.: Effects of molecular structure on aerosol yields from OH radical-initiated reactions of linear, branched, and cyclic alkanes in the presence of NOx, Environ. Sci. Technol., 43(7), 2328–2334, doi:10.1021/es803389s, 2009.

Nannoolal, Y., Rarey, J. and Ramjugernath, D.: Estimation of pure component properties, Fluid Phase Equilib., 269(1–2), 117–133, doi:10.1016/j.fluid.2008.04.020, 2008.

Ng, N. L., Chhabra, P. S., Chan, a. W. H., Surratt, J. D., Kroll, J. H., Kwan, a. J., McCabe, D. C., Wennberg, P. O., Sorooshian, A., Murphy, S. M., Dalleska, N. F., Flagan, R. C. and Seinfeld, J. H.: Effect of NOx level on secondary organic aerosol (SOA) formation from the photooxidation of terpenes, Atmos. Chem. Phys., 7(4), 5159–5174, doi:10.5194/acpd-7-10131-2007, 2007.

Shrivastava, M., Cappa, C. D., Fan, J., Goldstein, A. H., Guenther, A. B., Jimenez, J. L., Kuang, C., Laskin, A., Martin, S. T., Ng, N. L., Petaja, T., Pierce, J. R., Rasch, P.

J., Roldin, P., Seinfeld, J. H., Shilling, J., Smith, J. N., Thornton, J. A., Volkamer, R., Wang, J., Worsnop, D. R., Zaveri, R. A., Zelenyuk, A. and Zhang, Q.: Recent advances in understanding secondary organic aerosol: Implications for global climate forcing, Rev. Geophys., 55(2), 509–559, doi:10.1002/2016RG000540, 2017.

Stone, D., Whalley, L. K. and Heard, D. E.: Tropospheric OH and HO2 radicals: field measurements and model comparisons, Chem. Soc. Rev., 41(19), 6348, doi:10.1039/c2cs35140d, 2012.
* * *